# Hepatitis B Virus Exploits ERGIC-53 in Conjunction with COPII to Exit Cells

**DOI:** 10.3390/cells9081889

**Published:** 2020-08-12

**Authors:** Lisa Zeyen, Tatjana Döring, Reinhild Prange

**Affiliations:** Institute for Virology, University Medical Center of the Johannes Gutenberg University Mainz, Augustusplatz, D-55131 Mainz, Germany; zeyen@uni-mainz.de (L.Z.); doeringt@uni-mainz.de (T.D.)

**Keywords:** HBV, HBV assembly, HBV egress, COPII, Sec24A, ERGIC-53, LMAN-1, N-glycosylation

## Abstract

Several decades after its discovery, the hepatitis B virus (HBV) still displays one of the most successful pathogens in human populations worldwide. The identification and characterization of interactions between cellular and pathogenic components are essential for the development of antiviral treatments. Due to its small-sized genome, HBV highly depends on cellular functions to produce and export progeny particles. Deploying biochemical-silencing methods and molecular interaction studies in HBV-expressing liver cells, we herein identified the cellular ERGIC-53, a high-mannose-specific lectin, and distinct components of the endoplasmic reticulum (ER) export machinery COPII as crucial factors of viral trafficking and egress. Whereas the COPII subunits Sec24A, Sec23B and Sar1 are needed for both viral and subviral HBV particle exit, ERGIC-53 appears as an exclusive element of viral particle propagation, therefore interacting with the N146-glycan of the HBV envelope in a productive manner. Cell-imaging studies pointed to ER-derived, subcellular compartments where HBV assembly initiates. Moreover, our findings provide evidence that HBV exploits the functions of ERGIC-53 and Sec24A after the envelopment of nucleocapsids at these compartments in conjunction with endosomal sorting complexes required for transport (ESCRT) components. These data reveal novel insights into HBV assembly and trafficking, illustrating therapeutic prospects for intervening with the viral life cycle.

## 1. Introduction

The hepatitis B virus (HBV) is one of the most successful human pathogens and remains a major health threat for the human population worldwide. About 257 million people are persistently infected, facing a high risk to develop liver cirrhosis and hepatocellular carcinoma [1]. Current treatments can control HBV replication without eliminating HBV infection. To develop new, curative therapies for chronic HBV carriers, an improved understanding of the viral life cycle is essential. Like all viruses, HBV needs to exploit the molecular machinery of a cell to multiply [2,3]. Hijacking host cell processes applies in particular to HBV, as its DNA genome with about 3 kb in size is one of the smallest viral genomes known [4,5].

HBV is a hepatotropic, enveloped virus that replicates by protein-primed reverse transcription. Its genome merely encompasses four overlapping open reading frames (ORF) encoding the polymerase/reverse transcriptase (Pol), the capsid-forming core protein and its cognate secretory precore protein, as well as the regulatory X protein and three related envelope proteins [4,5,6]. The envelope proteins accomplish virion attachment to hepatocytes via binding to the liver cell-specific sodium taurocholate cotransporting polypeptide (NTCP) [7]. After HBV uptake and disassembly, the viral nucleocapsid (NC) is delivered to the nucleus where the partially double-stranded viral genome is complemented into a covalently closed circular DNA. This replication intermediate serves as a template for viral RNA transcription. The formation of progeny virions begins with the assembly of the icosahedral NC that selectively incorporates the viral pregenomic RNA (pgRNA) together with the viral polymerase, thereby warranting the synthesis of the progeny DNA genome through reverse transcription of the pgRNA. Mature NCs can then be enclosed by the viral envelope, composed of cellular lipids and viral glycoproteins [2,4,5,6].

The three envelope proteins, the small (S), middle (M) and large (L) proteins, are expressed by a single ORF, utilizing three different start codons and a common stop codon. Consequently, the amino acid sequence of S (or the S domain) is repeated at the C-termini of M and L that carry the additional preS2 domain or preS1 plus preS2 domains, respectively. All three proteins are synthesized as multi-spanning transmembrane proteins at the endoplasmic reticulum (ER) membrane (see also Figure 4A). Despite their structural similarities, the envelope proteins massively differ in their functions regarding the viral life cycle. S is the major component of the envelope and builds up its scaffold, while M is dispensable for virus formation [8]. The L protein is essential for viral particle production and infectivity, as it mediates capsid recruitment to virion budding sites and NTCP-receptor binding, which is enabled by its dual transmembrane topology [7,9,10]. Within the ER, all three envelope proteins acquire N-linked glycans attached to an asparagine residue (N146) located within a hydrophilic, ER-luminally disposed loop of the S domain. Intriguingly, N-linked glycosylation is essential for the outcome of an HBV infection, as an inhibition of this modification blocks virion egress [11,12,13,14,15,16]. Its significance is substantiated by the strict conservation of N146-linked N-glycosylation in all eight HBV genotypes [17] and even in related hepadnavirus family members, like the woolly monkey hepatitis B virus [18] and the woodchuck hepatitis B virus [19]. The functional role of the HBV-specific N-glycans is less clear.

Similarly, the HBV budding site and export route out of the hepatocyte remain unidentified. Several studies revealed that HBV virion budding occurs at intracellular membranes and depends on functions of the endosomal sorting complexes required for transport (ESCRT) machinery [20,21,22,23,24]. This machine accomplishes the generation of intraluminal vesicles that bud away from the cytosol into the lumen of multivesicular bodies (MVBs). In addition, ESCRTs were also found to be involved in topologically equivalent membrane remodeling events, notably the facilitation of enveloped virus budding and cytokinetic abscission [25]. The machine comprises the heteromeric ESCRT-0, -I, -II and –III complexes, along with the Vps4 ATPase, which function in a sequential manner.

In addition to the mature virions, HBV produces and releases subviral particle types—in particular, noninfectious, spherical empty envelope subviral particles (SVPs) [26]. These particles, also referred to as hepatitis B surface antigen (HBsAg) particles, are secreted in extreme surplus compared to the infectious HBV particles and are thought to act as decoys to the immune system [26,27]. The viral S envelope protein alone is sufficient for the production of spherical SVPs with diameters of about 22 nm that are formed by self-assembly of about 100 S molecules in the ER-Golgi intermediate compartment (ERGIC) and are released via the constitutive pathway of secretion [28,29]. Remarkably, the budding and secretion of spherical SVPs does neither require N146-linked glycans [11,13,14] nor ESCRTs [21,23], implicating that HBV exploits distinct cellular pathways and host factors to release its particle types. Besides spherical SVPs, HBV-infected hepatocytes also secrete filamentous SVPs with 22-nm diameters and variable lengths. Unlike spheres, filaments are characterized by a large content of the L envelope protein and exploit ESCRT functions for cell exit [21,23].

In the search for host factors involved in HBV particle-type exports, we recently employed yeast-based proteomics and identified Sec24A as an interaction partner of the HBV envelope S domain [29]. Sec24 is the cargo adaptor of the coat protein complex II (COPII), a vesicular transport machinery that moves selected secretory cargos from the ER to the Golgi apparatus. Sec24 acts in conjunction with the GTPase Sar1 and Sec23 to build the inner coat layer of COPII, while the Sec13/Sec31 complex forms the outer coat layer, thereby promoting the vesicle budding of cargo-loaded COPII vesicles [30,31]. Mapping analyses additionally showed that the Sec24A isoform binds to a di-arginine motif of the HBV S domain, a novel ER export code, in a productive manner. A blockage of the COPII/HBV crosstalk, created by interference with this binding motif, inhibited the ER exit of S and, thus, the secretion of the spherical SVPs [29].

In view of the supposed differences in host cell exits, we here examined the role of the COPII machinery in HBV viral particle trafficking and egress. We found that HBV virion release exhibited the same COPII selectivity as SVP secretion, as it strictly required the action of Sec24A and Sec23B. Moreover, we herein discovered ERGIC-53 (ER-Golgi intermediate compartment 53 protein; also known as lectin mannose-binding 1 protein, LMAN-1) as a selective interaction partner of HBV that recognizes the viral envelope in an N146-linked glycan-dependent manner. The silencing of ERGIC-53 inhibited HBV virion egress without affecting HBV spherical SVP release, implicating that it may serve as a device to segregate intracellular trafficking routes of HBV particle types. Beyond that, we show that the viral envelope captures nucleocapsids in ER-associated compartments for envelopment concomitant with ESCRT recruitment.

## 2. Materials and Methods

### 2.1. Expression Constructs

The plasmid pHBV carries a 1.1 mer of the HBV genome (nt 2922 to 84, derived from genotype D, GenBank™ accession number V01460) under the transcriptional control of the human metallothionein (hMT) IIA promoter in a pUC19 backbone [32]. To prevent expression of the L, M and S envelope proteins (HBV.Env^minus^), their translational start codons were inactivated by mutagenesis, as described [33]. In the HBV.L^minus^ construct, only the L-specific start codon was substituted by a threonine residue [33]. To ablate the core expression from the HBV replicon (HBV.C^minus^), a stop codon at triplet 38 of the core gene was created [34]. The mutant HBV.Pol^minus^ replicon carries an aspartic acid-to-histidine substitution at amino acid (aa) position 540 of the HBV polymerase and has been shown to be defective in reverse transcriptase activity [32]. For trans-complementation studies of the HBV.C^minus^ replicon, the vector pNI2.C containing the HBV core gene preceded by the hMT IIa promoter was used [35]. The core mutant C.K96A, defective in capsid envelopment, has been described [34]. The plasmid pT7HB2.7, which directs the synthesis of all three HBV envelope proteins under their authentic promoters, was kindly provided by C. Sureau (INTS, Paris, France) [13] and used for trans-complementation of the HBV.Env^minus^ replicon. To inactivate the N-glycan acceptor site within the S domain of the three envelope proteins, the asparagine residue at aa position 146 (N146) was replaced by a glutamic acid residue (Env.N146D), as described [36]. For the sole expression of the S protein under the control of the hMT IIa promoter, plasmid pNI2.S.HA carrying the S gene with a C-terminal-fused haemagglutinin (HA) tag was used [10]. Where indicated, HBV replication was studied in cells transfected with pCEP4∆CMV∆SV40/1.1 × HBV (termed pHBV* herein), which carries the 1.1 mer of the HBV genome in a modified pCEP4 plasmid backbone [22]. The plasmid pEYFP-ER encodes an enhanced yellow fluorescent protein (EYFP) with an N-terminal-fused ER-targeting sequence of calreticulin plus a C-terminal-fused ER retrieval sequence and was purchased from Clontech (Clontech, Mountain View, CA, USA)**.** The vector pGRASP65-GFP (green fluorescent protein) was kindly provided by E. Hildt (PEI, Langen, Germany) and encodes a fusion protein of GRASP65, a peripheral membrane protein that is localized to the cis-Golgi, and of the green fluorescent protein (GFP). The plasmid pDsRed-Golgi carries the DsRed-coding sequence fused to the N-terminal 81 aa of human β 1,4-galactosyltransferase, which targets the fusion protein to the trans-medial region of the Golgi (Clontech, Mountain View, CA, USA). The expression vector encoding human Vps4A with a C-terminal EGFP tag was generously provided by W. Sundquist (University of Utah, Salt Lake Cit, UT, USA). The vector pcDNA3.1(+)-ERGIC-53 carrying the human *ERGIC-53* gene (NM_005570.3) under the control of the CMV promoter was purchased from GenScript (GenScript Biotech, Leiden, Netherlands). For tagging with the Myc epitope, the aa sequence 28–37 of ERGIC-53 was mutated to EQKLISEEDL by Q5^®^ Site-Directed Mutagenesis (New England Biolabs, Ipswich, MA, USA) with the oligonucleotides 5′- CCGAGGAGGACCTCCTGGTGCAGAGCGACGG-3′ and 5′- AGATTAATTTTTGCTCGAAACGGCGATGTGGC-3′.

### 2.2. SiRNAs, Cell Culture and Transfection

For transient expression analyses, the human hepatocellular carcinoma cell line HuH-7 was used that was obtained by the European Collection of Authenticated Cell Cultures (http://cellbank.nibiohn.go.jp/english/). This cell line is not susceptible to HBV infection, because it expresses very low levels of the NTCP receptor and is therefore a useful model to study the production and release of the virus rather than infection [7]. Cells were cultured in Dulbecco’s modified Eagle’s medium (Thermo Fisher Scientific, Waltham, MA, USA) supplemented with 10% fetal bovine serum and 5-µg/mL ciprofloxacin (Fresenius Kabi, Bad Homburg, Germany). Transfections of cells with plasmid DNAs were performed with Lipofectamine^™^ Plus (Thermo Fisher Scientific, Waltham, MA, USA). For depletion of EAP20 (NM_032353.3), Sar1A (NM_01142648), Sar1B (NM_016103.3), Sec23A (NM_006364.4), Sec23B (NM_006363), Sec24A (NM_021982) or Sec24B (NM_006323), single siRNA duplexes or siGENOME SMARTpool RNAs (Dharmacon, Lafayette, CO, USA) were used as described [22,29]. To silence the expression of ERGIC-53 (NM_005570), a siRNA (5′-GGACAGAAUCGUAUUCAUC-3′) targeting nucleotide positions 1009–1027 was obtained from Dharmacon (Dharmacon, Lafayette, CO, USA). The specificity and efficacy of this siRNA has been approved in independent studies [37,38,39]. A control siRNA with no known homology to mammalian genes was purchased from Qiagen (Qiagen, Hilden, Germany). For combined transfection, HuH-7 cells were first transfected with siRNAs by using the Lipofectamine^™^ RNAiMAX transfection reagent (Thermo Fisher Scientific, Waltham, MA, USA). In a typical experiment, 3 × 10^5^ cells per well of a 12-well plate were transfected with a final concentration of 20-nM siRNA, according to the protocol of the supplier. After 48 to 72 h, cells were retransfected with 2-µg plasmid DNA, and transfected cells were harvested after an additional 48 to 72 h, as indicated in the text. For drug treatment, cells were incubated with 1-µM brefeldin A (BFA; Sigma-Aldrich, St. Louis, MO, USA) for 2 h at 37 °C.

### 2.3. Antibodies

Polyclonal antisera against recombinant native (K45) or denatured (K46) HBV core particles were raised in rabbits, as described [35]. In addition, commercially available polyclonal rabbit (B0586; Dako, Carpinteria, CA, USA) or monoclonal mouse (3HB17; HyTest, Turku, Finland) antisera against the core antigen or capsid particle, respectively, were used. For immunodetection of the HBV L protein, a rabbit antibody raised against a recombinant peptide encoding aa 1–42 of L fused to glutathione *S*-transferase (K1350; Eurogentec, Liège, Belgium) or the mouse antibodies HepB preS1 (sc-57761; Santa Cruz, CA, USA) and MA18/7 (D. Glebe, University of Giessen, Giessen, Germany), recognizing L-specific epitopes, were employed. For immunoprecipitation (IP) of all three HBV envelope proteins, a rabbit polyclonal antiserum (K38) against the S domain was used [33]. Antibodies obtained from Covance (Covance, HISS Diagnostics, Freiburg im Breisgau, Germany) were mouse anti-HA (16B12) and anti-Myc (9E10). The mouse anti-EAP20 (VPS25: sc-271648) antibody was purchased from Santa Cruz (Santa Cruz, CA, USA), while the mouse anti-ERGIC-53 (OTI1A8) antibody was from Enzo (Enzo Life Sciences, Farmingdale, NY, USA). Antibodies obtained from Sigma-Aldrich (Sigma-Aldrich, St. Louis, MO, USA) were rabbit anti-ERGIC-53 (E1031), rabbit anti-Sec24A (HPA056825), anti-β-actin (AC15) and anti-α-tubulin (B-5-1-2). Peroxidase-labeled secondary antibodies were obtained from Dianova (Hamburg, Germany), and fluorophore-labeled antibodies were purchased from Thermo Fisher Scientific (Thermo Fisher Scientific, Waltham, MA, USA).

### 2.4. Quantitative Reverse Transcription PCR (qRT-PCR) Analysis

Total mRNAs were isolated from cells using the peqGold TriFast (Peqlab Biotechnologie, Darmstadt, Germany) and the Direct-zol^™^ RNA MiniPrep kit (Zymo Research, Irvine, CA, USA). The mRNA was digested with 5 U RNase-free, recombinant DNase I (Roche Diagnostics, Rotkreuz, Switzerland), and cDNA synthesis was performed by using the qScript cDNA Synthesis Kit (Quanta Biosciences, Beverly, MA, USA). qRT-PCR reactions were conducted as described [40]. For data analysis, the comparative cycle threshold method (C*_T_*) was used, and data were reported as the fold change normalized to an endogenous reference gene (β-actin). To quantify the ERGIC-53 transcript levels, the primers 5′- CAGATCAAATTCGAGTAGCACCA-3′ and 5′-AATATTCCAACACCATTCCACAGA-3′ were used. All other primer pairs have been described previously [29].

### 2.5. Subviral Particle Analysis

To probe for subviral particle production, an SVP release assay was used, as described [29]. Briefly, cells were lysed for 20 min on ice, and lysates were centrifuged for 5 min at 15,000× *g* and 4 °C. To analyze the secretion of SVPs from transfected cells, clarified culture medium was concentrated by ultracentrifugation through a 20% (*w/v*) sucrose cushion (4 h at 100,000× *g* and 4 °C), and samples were subjected to Western blot (WB) analysis.

### 2.6. Viral Particle Analysis

The production and release of HBV particles were determined by a TaqMan^®^ chemistry-based real-time PCR, as described [22]. Briefly, cell lysates and supernatants were harvested, and intracellular nucleocapsids and extracellular virions were isolated by immunomagnetic separation using PureProteome^™^ Protein G Magnetic Beads (Millipore, Billerica, MA, USA) coated with capsid (K45)- and envelope-specific (K38 and K1350) antibodies, respectively. After isolation of the viral DNA, PCR analyses were performed with a 7500 Real-Time PCR System and Sequence Detection Software 4.0 (Applied Biosystems, Foster, CA, USA). Dependent on the transfected HBV replicon constructs, simplex or multiplex PCR analysis were conducted, with the latter using two primer/probe sets targeting either the HBV genome or the hygromycin resistance gene of the pCEP plasmid backbone, as described [22]. Where indicated, cells were lysed by osmotic shock in a hypotonic lysis buffer (10-mM Tris-HCl, pH 7.5, 10-mM NaCl and 1.5-mM MgCl_2_) for 15 min on ice and three subsequent freeze-thaw cycles (using liquid nitrogen to freeze and a 37 °C water bath to thaw). Lysates were supplemented with NaCl to a final concentration of 150 mM and centrifuged for 30 min at 15,000× *g* and 4 °C. Intracellular enveloped viral particles were immunoisolated by an envelope-specific IP in the absence of detergents and assayed by PCR.

### 2.7. Endoglycosidase H (EndoH) Treatment

For EndoH digestion, cell supernatants were concentrated by ultracentrifugation through a 20% (*w/v*) sucrose cushion (4 h at 100,000× *g* and 4 °C), and pellets were suspended in Tris-buffered saline TBS buffer (TBS; 50-mM Tris-HCl, pH 7.5, and 150-mM NaCl). Cells were lysed in TBS supplemented with 0.5% Nonidet P-40 (NP-40), and lysates were centrifuged for 5 min at 15,000× *g* and 4 °C. After the addition of 10 × glycoprotein denaturing buffer (New England Biolabs, Ipswich, MA, USA), samples were heated at 100 °C for 10 min. Next, samples were adjusted to 10 × GlycoBuffer 3 (New England Biolabs, Ipswich, MA, USA), divided in two portions and incubated for 1 h at 37 °C with or without 2-µL EndoH (New England Biolabs, Ipswich, MA, USA).

### 2.8. Coimmunoprecipitation (CO-IP) Assay

To probe for intracellular protein interactions, cells were either lysed with a 2% CHAPS solution dissolved in HBS (50-mM Hepes-KCl, pH 7.5, 200-mM NaCl and 20-mM N-ethylmaleimide) or with Triton buffer (50-mM Tris-HCl, pH 7.5, 150-mM NaCl, 1-mM CaCl_2_ and 1% Triton X-100) in the presence of 1 × protease inhibitor mixture for 20 min on ice. After centrifugation for 20 min at 15,000× g and 4 °C, lysates were immediately subjected to immunoprecipitation using tosyl-activated, superparamagnetic polystyrene beads (Dynabeads Sheep anti-rabbit or Dynabeads Sheep anti-mouse IgGs; Thermo Fisher Scientific, Waltham, MA, USA) that had been precoated with the anti-core (K45) or anti-L (57761) antibodies, as described [41]. After incubation for 3 h at 4 °C with agitation, the immune complexes were washed three times with either 0.5% CHAPS/HBS (CHAPS-lysed cells) or with 50-mM Tris-HCl, pH 7.5, 150-mM NaCl, 1-mM CaCl_2_ and 0.1% Triton X-100 (Triton-lysed cells).

### 2.9. Fluorescence Microscopy

For immunofluorescence (IF), cells grown on coverslips were fixed with 4% paraformaldehyde (PFA) for 10 min at room temperature and permeabilized with 0.2% Triton X-100 for 10 min. Cells were blocked in phosphate-buffered saline (PBS) containing 1% BSA, incubated with the indicated primary antibodies for 1 h at 37 °C, rinsed with PBS and then incubated with Alexa Fluor-tagged secondary antibodies for 1 h at 37 °C. Cell nuclei (DNA) were stained with Hoechst 33,342 (Sigma-Aldrich, St. Louis, MO, USA). Images were acquired separately for each channel using a Zeiss Axiovert 200 M microscope equipped with a Plan-Apochromat 100× (1.4 NA) and a Zeiss Axiocam digital camera (Zeiss, Oberkochen, Germany). AxioVision software 4.7.1 (Zeiss, Oberkochen, Germany) was used for merging pictures, and Tiffs were assembled into figures using Adobe Photoshop CS6. For quantitative colocalization analysis (QCA), digital photographs were quantitated using the AxioVision colocalization software (4.7.1; Zeiss, Oberkochen, Germany) by calculating the pixel intensity-based Pearson’s correlation coefficients (PCC) of randomly selected cells (25 cells per coverslip).

### 2.10. Statistics

Statistical differences between groups and statistical graphs were assessed with a two-tailed, unpaired *t*-test using Microsoft Office Excel 2016. Differences between groups were considered significant when the *p*-value was * *p* < 0.05 or ** *p* < 0.01.

## 3. Results

### 3.1. HBV L Resides within Markedly Crescent-Shaped ER-Associated Compartments

Unlike those of HBV subviral envelope particles (SVPs), intracellular trafficking pathways of the viral envelope are poorly understood. Unlike those of HBV spherical subviral envelope particles (SVPs) formed by S alone, intracellular trafficking pathways of the L are poorly understood. To study HBV envelope transport through the cell secretory system, we used a transient replication system by transfecting HuH-7 cells with a replication-competent HBV replicon plasmid and performed IF analysis. For organelle-specific reporters, cells were cotransfected with either pYFP.ER, pYFP.GRASP65 or pDsRed.Golgi that encode autofluorescent marker proteins specific for the ER, cis Golgi or medial/trans Golgi, respectively. Three days post-transfection, cells were fixed with PFA, permeabilized and stained with antibodies against the L envelope protein. To locate the ER/Golgi intermediate compartment (ERGIC), cells were co-stained with antibodies against ERGIC-53, an approved marker of this organelle [42]. As shown in Figure 1A, L reproducibly appeared in a confined, crescent-shaped structure in the perinuclear region, partly emerging with dissections. For short, this structure will be termed CS herein. Worth mentioning, a similar pattern emerged when S domain-specific antibodies were used for staining (data not shown), indicating that the CS structure is specific for the viral envelope rather than for L. As would be expected, a similar pattern emerged when S domain-specific antibodies were used for staining (Figure 1A). To assess the impact of the L protein on the formation of the CS structure, comparative studies were performed with a mutant HBV.L^minus^ replicon construct that is defective in L protein synthesis [33]. In the absence of L, the S domain-specific antibody rendered a diffuse granular staining pattern of the M and S envelope proteins (Figure 1A), indicating that the CS structure is specific for L. As would be expected for a viral transmembrane envelope protein, the L-specific staining pattern clearly colocalized with the reticular structures labeled by the ER marker YFP.ER (Figure 1B). However, unlike many other viral envelope proteins synthesized within and transported through the secretory system, the L-specific CS structure neither colocalized with ERGIC- nor Golgi-specific areas (Figure 1B).

To investigate whether HBV might traverse post-ER compartments or not, we analyzed the nature of the N-glycans that are added to the asparagine acceptor site N146 of about half of the L molecules during their integration into the ER membrane. During glycoprotein transport through the secretory pathway, Golgi-residing enzymes process glycoproteins and modify N-linked mannose-rich oligosaccharides into complex glycans that are resistant to treatment with Endoglycosidase H (EndoH). By implication, an EndoH sensitivity of glycoproteins indicates the presence of high-mannose glycans and the absence of glycoprotein modification by Golgi-residing enzymes and vice versa. We therefore conducted EndoH digestions of intra- and extracellular HBV particles. As shown in Figure 1C, L is naturally synthesized in nonglycosylated p39 and single-glycosylated gp42 forms due to its partial N-glycosylation. Cell-associated L was entirely sensitive to EndoH, which converted the gp42 form to the p39 form. By contrast, extracellular L resisted EndoH digestion concomitant with an increased molecular weight of its gp42 form by about one kDa as a consequence of N-glycan processing. Accordingly, the viral envelope must have traversed the Golgi apparatus en route to the exterior. To account for the intracellular EndoH sensitivity of L together with its missing colocalization with ERGIC- and Golgi-specific markers, L might be blocked in a pre-ERGIC compartment for much of its time in the cell to match the rate-limiting step in viral particle assembly. Once budded, the viral particles appeared to be rapidly released out of the cell.

### 3.2. HBV L Recruits Core/Capsid to the Crescent-Shaped ER-Associated Compartments

To gain insights into HBV assembly/budding sites, we comparatively inspected the intracellular distribution of wt and mutant HBV replicon constructs on a single cell level. The mutant replicons were either defective in core synthesis (HBV.C^minus^), envelope synthesis (HBV.Env^minus^), L protein expression (HBV.L^minus^), reverse transcription (HBV.Pol^minus^) or in nucleocapsid envelopment (HBV.C.K96A). In HBV.C^minus^-expressing HuH-7 cells, L appeared in its typical CS structure, implicating that this feature is an intrinsic capacity of L, irrespective of an ongoing replication (Figure 2A). Upon ablation of the synthesis of all three envelope proteins, core yielded a diffuse staining dispersed throughout the cytoplasm with some nuclear labeling (Figure 2B). An almost identical distribution of core was observed, when only the expression of the L envelope protein was prevented (Figure 2C). In the presence of the viral envelope, the core staining pattern thoroughly changed, as it now accumulated in the L-specific CS structure (Figure 2D). A similar overlap and recruitment of core by L was observed when capsid- rather than core-specific antibodies were used for staining (Figure 2E). Even core/capsids defective in reverse transcription, but competent for pgRNA genome packaging [43], were found to extensively colocalize with the characteristic CS structure of L (Figure 2F). Conversely, when the HBV.C^minus^ replicon was trans-complemented with a core mutant that had been shown to be competent in nucleocapsid formation but defective in envelopment (HBV.C.K96A) [44], no recruitment of core could be detected (Figure 2G). To corroborate whether the spatial overlaps of the core and L-staining pattern relied on true protein interactions, coimmunoprecipitation (CO-IP) studies were performed. To this aim, cell lysates of transfected cells were subjected to a core-specific IP followed by L-specific WB. Thereby, wt core but not the envelopment-defective core.K96A mutant reproducibly brought down L (Figure 2H). Together, the cell imaging and biochemical data indicate that L on its own accumulated in the confined CS structure towards core/capsid particles were actively recruited, likely via a direct interaction between L and core. Since the interaction-defective core.K96A mutant failed to be attracted by L, the CS structure likely mirrors HBV interaction sites.

### 3.3. HBV Particle Egress Requires the COPII components Sec24A, Sec23B and Sar1

We previously showed that HBV spherical SVPs use the cellular COPII machinery for ER export and secretion [29]. Since intracellular trafficking pathways of HBV subviral and viral particles are supposed to differ [2,21,45], a common exploitation of COPII transport carriers is ambiguous. Therefore, we reasoned to investigate the role of COPII factors in the production and release of infectious HBV particles. For RNA interference (RNAi), HuH-7 cells were treated with siRNA duplexes for 48 h prior to transfection with the pHBV* replicon construct. After an additional 72 h, cell lysates and supernatants were harvested. The used siRNAs comprised control duplexes (siCon) and single or siRNA pools targeting Sec24A, Sec24B, Sec23A, Sec23B, Sar1A or Sar1B. Sec24 is the cargo adaptor protein of COPII and associates with Sec23 to form the inner coat layer, while the Sar1 GTPase recruits the Sec23/Sec24 dimer to the ER membrane. To monitor depletion, COPII-specific transcripts were measured by qRT-PCR, which revealed an almost complete knockdown (KD) of each targeted COPII component (Figure 3). Cell lysates were probed for the expression and stability profiles of the viral L envelope and core proteins by specific WBs. As shown in Figure 3, neither knockdown (KD) affected the intracellular steady-state levels of either of the two viral proteins. The formation and release of HBV particles were analyzed by a virion production (VP) assay. For this, intracellular NCs and extracellular virions were immunoprecipitated with capsid- or envelope-specific antibodies, respectively, followed by particle disruption and real-time, multiplex PCR quantification of the number of HBV progeny genomes. Neither RNAi had a substantial effect on the formation of intracellular, replication-competent NCs. However, upon the silencing of Sec24A and Sec23B, the amounts of extracellular virions decreased in a highly significant manner, as compared to siCon-treated cells (Figure 3). Conversely, the loss of the two closely related Sec24B [46] or Sec23A [47] isoforms had no inhibitory effects on virus production and release. This indicates that HBV strictly depends on the specific functions provided by Sec24A and Sec23B in the same manner as HBV SVPs. In case of the KDs of the two Sar1 isoforms, the inhibition of the HBV release was obvious, albeit less pronounced (Figure 3). Since the Sar1A and Sar1B isoforms share 89% sequence identity concomitant with overlapping functions [48], the individual loss of either isoform may be partly compensated by its paralog. Together, these results demonstrate that HBV viral particles, similar to spherical subviral particles, require the COPII machinery for exiting the host cell, involving functions of Sec24A, Sec23B and Sar1. Hence, a COPII-guided export out of the ER appears to be a mandatory step in HBV biology.

### 3.4. HBV L Colocalizes with ERGIC-53 upon ER Export Block

Given that the Sec24A isoform is the essential COPII component linked to the export of both the viral and subviral HBV envelope, we next investigated the fate of the ER-arrested envelopes due to Sec24A KD. To study spherical SVP trafficking, HuH-7 cells were transiently transfected with an expression construct encoding only the S envelope gene (HBV.S) (Figure 4A). Notably, this is an established approach to study exclusively the fate of subviral HBV spheres [29]. Consistent with our previous IF analysis [29] and for recapitulation herein, HBV.S showed a diffuse granular staining pattern typical for viral surface proteins that are synthesized at the ER (Figure 4B). Moreover, it largely colocalized with ERGIC-53-stained structures, implicating that HBV.S is transported out of the ER to the ERGIC in control cells (Figure 4B). Upon Sec24A inactivation, the staining pattern of HBV.S strongly changed, as it now appeared in dot-like structures, reminiscent of ER-arrested aggregates, that no longer coincided with ERGIC-53 signals (Figure 4B). Notably, under these conditions, the distribution of ERGIC-53 also altered. In normal cells, this protein predominantly localizes to the ERGIC compartment, as displayed by the given name ERGIC-53. It is a high mannose-specific lectin that cycles cargoes between the ER and the cis-Golgi through COPII and COPI-dependent pathways (see also Figure 5A) [49,50]. Upon immunostaining, it appeared in juxtanuclear, tubulovesicular clusters in control cells, which shifted to a perinuclear, ER-like structure in Sec24A KD cells (Figure 4B). Hence, the ER export of both HBV.S and ERGIC-53 is apparently impaired in Sec24A-silenced cells.

In contrast, the HBV L protein, synthesized in HBV-replicating cells, could not be traced within the ERGIC, as no colocalization between L and ERGIC-53 was detectable in the control cells (Figure 4C). In further difference to spherical SVPs, the naturally occurring structure of L was preserved in Sec24A KD cells without any signs of aggregation. Quite unexpectedly, in this setting, ERGIC-53 intensively colocalized with L, implicating that the lectin is seemingly entrapped by the L-specific CS structure if arrested in the ER. Similar staining experiments were done in cells depleted for the related Sec24B isoform. However, this intervention did not affect the typical juxtanuclear distribution of ERGIC-53 and, consequently, did not result in ERGIC-53/L colocalization (Figure 4C). We infer from these results that ERGIC-53 may be a preferred cargo of Sec24A rather than of Sec24B. If arrested in the ER, ERGIC-53 tightly associated with the HBV L envelope protein either by chance or for action.

### 3.5. HBV Particle Egress Requires ERGIC-53, while HBV Spherical Subviral Particle Egress Does Not

To distinguish between these possibilities, we probed for a functional role of ERGIC-53 in HBV propagation. HuH-7 cells were treated with control RNA duplexes or a single ERGIC-53-specific siRNA prior to transfection with the HBV replicon (pHBV*). The analysis of ERGIC-53-specific transcript and protein levels revealed an almost complete KD without any signs of cell cytotoxicity, as evidenced by anti-tubulin immunoblotting of the same cell lysates (Figure 5B). The loss of ERGIC-53 did not affect the L or core protein expression, nor the N-glycosylation of L (Figure 5B). Importantly, the virus production assay revealed that the silencing of ERGIC-53 significantly reduced the amounts of extracellular virions without altering the intracellular nucleocapsid assembly (Figure 5B). Hence, these results indicate that the lectin plays an essential role in HBV envelopment and transport out of the cell.

Given the important role of Sec24A in guiding ERGIC-53 ER export and its involvement in HBV subviral and viral envelope trafficking, we next investigated whether ERGIC-53 may likewise contribute to HBV spherical SVP secretion. For this, we employed an SVP release assay and transiently transfected HuH-7 cells with an expression construct encoding only the S envelope gene (HBV.S) with a C-terminally fused HA-tag and examined the intra- and extracellular HBV.S levels by HA-specific WB. For RNAi, cells were treated with control or ERGIC-53-specific RNA duplexes (72 h) prior to transfection with HBV.S (48 h). To control the accuracy of the SVP release assay, cells were depleted for Sec24A, an intervention known to block SVP secretion [29]. Lysates and supernatants were harvested, and SVPs present in the supernatants were collected by ultracentrifugation through sucrose cushions. Immunoblotting of lysates with anti-ERGIC-53 and anti-Sec24A antibodies demonstrated efficient depletions without any signs of codepletions (Figure 5C). Consistent with our previous results [29], Sec24A silencing strongly inhibited the secretion of HBV.S as compared to the control cells and concomitantly increased the intracellular S.HA levels (Figure 5C). In contrast, the KD of ERGIC-53 neither interfered with the synthesis of the nonglycosylated and glycosylated HBV.S forms nor with their secretion (Figure 5C). Together, these data indicate that the export routes for HBV viral and spherical subviral particles dramatically differ in their requirements for ERGIC-53.

### 3.6. Blocking N146-linked Glycosylation Inhibits HBV Egress without Affecting Envelope/Core Interactions

Since ERGIC-53 acts as a high mannose-specific lectin, we next focused on the HBV-specific N-glycans. A number of studies have shown that HBV release requires sugar moieties linked to the N-glycan acceptor N146 of the envelope proteins [11,12,13,14,15,16]. To ascertain the importance of the N146-linked glycans in our experimental settings, we disabled N-glycosylation by creating an asparagine-to-aspartic acid substitution in the HBV envelope expression vector (Env.N146D). This vector was then used for trans-complementation of the envelope-defective replicon HBV.Env^minus^, followed by WB and VP assays. Consistent with the literature [11,12,13,14,15], the Env.N146D mutant was exclusively synthesized in the nonglycosylated p39 form and, importantly, inhibited the HBV virion release (Figure 6A). Previous studies have attributed the virion release defect of Env.N146D to possible structural alterations of L that might prevent its interaction with the viral capsid [13]. To address this point, we performed CO-IP and CO-IF experiments of HBV-replicating cells expressing either the wt Env or mutant Env.N146D proteins. Neither assay rendered any evidence for a disturbed L/core interaction, as core efficiently coprecipitated Env.N146D (Figure 6B) and thoroughly colocalized with L.N146D in the characteristic CS structure (Figure 6C). We thus infer that the N146-linked N-glycan plays a pivotal role in HBV egress apart from envelope/core interactions.

### 3.7. HBV L Interacts with ERGIC-53 in a N146-Glycan-Dependent Manner

Accordingly, we next investigated whether there might be physical links between the viral N-glycan and cellular ERGIC-53. As above, HuH-7 cells were depleted for Sec24A in order to arrest ERGIC-53 within the ER, following a retransfection with the wt replicon construct or the envelope-defective HBV.Env^minus^ replicon trans-complemented with Env.N146D. In control cells, neither the wt L nor the L.N146D mutant exhibited any colocalization with the tubulovesicular clusters stained with anti-ERGIC-53 (Figure 7). Upon depletion of Sec24A, the ERGIC-53-specific labeling pattern changed and again appeared in ER-like reticular areas. Importantly, this pattern strongly coincided with the CS structure of wt L (Figure 7A) but poorly with the CS structure of L.N146D (Figure 7B). To corroborate these findings, CO-IP analyses were performed. Since ERGIC-53/cargo interactions are known to be transient and short-lived [42,50], we here used the ectopic expression of a Myc-tagged ERGIC-53 construct together with the wt or mutant HBV replicons. As shown in Figure 7C, wt L, but not L.N146D, efficiently coprecipitated ERGIC-53. Together, these results point to a productive interaction between L and ERGIC-53 that is primary mediated by the N146-linked glycan.

### 3.8. HBV Envelopment of Nucleocapsids Occurs prior to the Actions of ERGIC-53 and Sec24A

To narrow down the roles of ERGIC-53 and Sec24A in the HBV´s life cycle, colocalization studies were performed in RNAi-treated HBV-replicating cells. Aside, cells were treated with BFA, an established fungal compound that inhibits protein trafficking and secretion in mammalian cells via the collapsing of ER exit sites and, hence, of ER export [51]. The double-staining with anti-L and anti-core antibodies showed that neither the loss of ERGIC-53 nor of Sec24A hindered the recruitment of core to the typical CS structure of L (Figure 8A), indicating that both host factors are dispensable for L/core contact formation. Similar results were obtained in BFA-treated cells (Figure 8A), implicating that core is recruited by L prior to ER exit. To test whether ERGIC-53 and Sec24A might play a role in HBV assembly/budding or trafficking, we modified the VP assay in such that we screened for virions in cell extracts. In order to prevent the destruction of the viral envelope during lysis, cells were mechanistically disrupted. Extracts were next subjected to an L-specific IP in the absence of detergents, followed by particle disruption and PCR quantification of the number of HBV progeny genomes (Figure 8B). To assure the IP efficiency under these conditions, extracts precipitated in the same manner were analyzed by L-specific immunoblotting, which demonstrated effective and comparable precipitations of L (Figure 8C). Noteworthy, the pull-downed L pool may comprise both viral particles and subviral filamentous SVPs. Even so, filaments lack nucleocapsids and viral genomes and are not detectable by PCR analysis. In respect thereof, the PCR data, shown in Figure 8B, demonstrated that the depletion of either ERGIC-53 or Sec24A led to a significant increase of intracellular virions as compared to the control cells. This indicates a blockage of viral transport more than viral assembly.

### 3.9. HBV Recruits the ESCRT-specific Vps4 ATPase to the Crescent-Shaped ER-Associated Compartments

Several studies have shown that HBV budding and egress require functions of the ESCRT complex, a membrane scission machinery operating at several cellular membranes [20,21,22,23,24]. To examine whether the ER-related, core-recruiting CS structure of L might have a link to ESCRT, we inspected the intracellular distribution of the Vps4 ATPase, the terminal ESCRT subunit [25], in HBV-replicating and control HuH-7 cells. To this aim, cells were transfected with an EGFP-tagged version of Vps4A together with either empty plasmid DNA or the HBV replicon and processed for IF analysis. In control cells, Vps4A was distributed throughout the cytoplasm, along with some nuclear localization (Figure 9A). Conversely, in HBV-positive cells, the distribution of Vps4A changed in such that it now additionally appeared in the CS structure reminiscent for HBV (Figure 9A). The apparent recruitment of Vps4A to the L/core complex suggested an ESCRT intervention occurring at the ER-related CS structure. To further investigate the spatial and temporal interconnections between HBV and the ESCRT cascade, we reasoned to study the N-glycan pattern of L in order to learn whether the HBV budding driving activity of ESCRT occurred before or after its acquisition of EndoH resistance. For ESCRT inactivation in HBV-replicating HuH-7 cells, we silenced the expression of EAP20, one subunit of the heterotetrameric ESCRT-II complex. Consistent with our previous report [22], this led to a substantial decline of extracellular virion amounts (Figure 9B). In parallel, cellular lysates were mock-treated or digested with EndoH and assayed by L-specific WB. Upon ESCRT-II inactivation, cell-associated L displayed the same EndoH sensitivity as observed in siCon-treated cells (Figure 9B), implicating that EAP20 is acting prior to L arrival at the cis/medial Golgi complex.

## 4. Discussion

One important outcome of this work was the finding that HBV not only uses the ER membrane as a platform for protein synthesis but additionally subverts the ER export machinery for viral propagation. We herein identified ERGIC-53, a lectin-like cargo receptor within the early exocytic pathway, along with distinct COPII components as critical host factors guiding the egress of HBV viral particles.

Upon the infection of liver cells, HBV does not only produce enveloped progeny virions but, also, empty envelope subviral particles (SVPs) that are secreted in large excess compared to virions and are thought to neutralize the host immune response against infectious HBV [26,27,45]. By analyzing spherical SVP trafficking through the cell secretory pathway, we previously found that the pathogen usurps distinct components of the COPII vesicle-forming machinery for spherical SVP host cell exit [29]. As there is evidence suggesting that the trafficking and export routes of HBV spherical subviral and viral particles may differ [21,23], we recapitulated COPII interference experiments in HBV-replicating HuH-7 liver cell cultures. Thereby, we observed that HBV virion release exhibited the same COPII selectivity as SVP secretion, as it required a functioning Sar1 GTPase, along with the Sec24A and Sec23B isoforms. Despite the massive overproduction of SVPs, efficient HBV secretion was shown to be maintained upon the mutational ablation of the concomitant SVP formation [52]. We thus infer that the viral envelope per se requires COPII-guided export mechanisms out of the ER.

Surprisingly, however, we failed to detect the HBV envelope beyond the ER, as it colocalized neither with ERGIC- nor Golgi-specific structures. Rather, the envelope accumulated in perinuclear ER regions in the form of a confined, crescent-shaped structure (CS). Since this structure was preserved even while interfering with the ER exit, provoked by Sec24A silencing or BFA treatment, the HBV envelope appears to be blocked in a pre-ERGIC compartment for much of its time in the cell. In contrast, the HBV subviral envelope has been shown to be transported out of the ER to the ERGIC where SVP assembly takes place. Hence, although both envelope types are dependent on COPII-guided transport events, the underlying mechanisms may not be thoroughly identical.

In search for possible differences, we herein discovered the cellular ERGIC-53 lectin to be crucial for HBV viral particle release but dispensable for HBV spherical SVP secretion. ERGIC-53 is an ER export receptor for a limited number of glycoproteins and cycles between the ER, the ERGIC and the cis-Golgi through COPII and COPI-dependent pathways [42,49,50,53]. Its cytosolic tail contains an ER exit motif that binds to Sec24, the COPII cargo adaptor for anterograde transport, while an adjacent motif interacts with COPI to mediate the retrograde transport [50]. In-line with this, our IF studies showed that the trafficking of ERGIC-53, i.e., its export out of the ER was impaired upon the silencing of Sec24A. Mammalian cells contain four Sec24 isoforms that can be structurally and functionally divided into two subclasses, Sec24A/B and Sec24C/D. Interestingly, we found that the closely related Sec24B isoform failed to compensate for the loss of Sec24A and did not interfere with proper ERGIC-53 trafficking. Consistent with this, previous in vitro COPII budding assays and peptide-binding analyses revealed that all four Sec24 isoforms can, in principle, recognize ERGIC-53, though Sec24A turned out to be the most important isoform guiding the anterograde trafficking of the lectin [46,54]. Together, our results implicate that both the HBV envelope and cellular ERGIC-53 are selective clients of the Sec24A isoform.

Our observation that ER-trapped ERGIC-53 intensively colocalized with the HBV envelope provided a starting point for functional studies. The RNAi-mediated silencing of ERGIC-53 profoundly inhibited HBV egress without affecting the proper envelope protein synthesis, viral replication and nucleocapsid formation. Our finding that even the envelopment of viral nucleocapsids were not compromised in ERGIC-53-silenced cells strongly points to a role of the lectin in HBV trafficking rather than assembly. To do so, ERGIC-53 associates with the viral L glycoprotein, as evidenced by CO-IF and CO-IP studies. Conversely, the trafficking and secretion of HBV spherical SVP release did not require ERGIC-53. Amongst others, the HBV viral and subviral envelopes differ in their needs of N-glycans added to the N146 acceptor site. Several studies have shown that an inhibition of N-glycosylation evoked by either the mutational inactivation of N146 or pharmaceutical interventions blocked the virion release, which we confirmed herein. Under the same conditions, the spherical SVP secretion preceded normally [11,13,14]. The apparent correlation between N-glycan and ERGIC-53 requirements and vice versa implicate that the viral, but not the subviral, envelope is a client of the lectin. ERGIC-53 may thus serve as a pivotal device to segregate the maturation pathways of the HBV viral and subviral particles.

Although mammalian cells contain vast amounts of glycoproteins, only a subset of polypeptides are trafficked in an ERGIC-53-dependent manner. These include mostly soluble glycoproteins, like blood clotting factor V (FV) and factor VIII (FVIII), cathepsin C, cathepsin Z, α1-antitrypsin and matrix metalloproteinase-9 [37,55,56,57]. Its restricted clientele may explain why the loss of ERGIC-53 function is well-tolerated both in vitro and in vivo [55,58,59]. Merely, mutations in ERGIC-53 lead to the genetic bleeding disorder-combined deficiency of FV and FVIII, a rare autosomal recessive disease [58]. To account for the amazing small number of cargoes, interaction analyses revealed that ERGIC-53 often recognizes a combined oligosaccharide/peptide structure that may limit its repertoire of cargoes [60,61,62]. For example, its interaction with cathepsin Z is mediated by a β-hairpin loop next to the N-linked carbohydrate [59]. A carbohydrate- plus conformation-dependent recognition by ERGIC-53 may likewise apply to the HBV envelope types. Despite sharing the same primary sequences, the configuration of the L-specific S domain within the viral envelope may differ from the S protein within the subviral envelope. In favor of this view, L is known to exhibit a dual transmembrane topology [10,63] that may affect the signature recognized by ERGIC-53. Even so, based on the results obtained with the N-glycosylation-defective HBV envelope mutant, the glycan code dominates the protein code.

Thus far, there is one other report demonstrating a direct exploitation of ERGIC-53 by viruses [39]. Several highly pathogenic RNA viruses, including arenavirus, coronavirus and filovirus, have been shown to take over ERGIC-53 for propagation. Similar to HBV, the envelope glycoproteins of these viruses associate with ERGIC-53 in a productive manner [39]. However, dissimilar to HBV, loss of the ERGIC-53 function did not compromise arenavirus glycoprotein trafficking or particle release but impaired viral infectiousness. As an underlying mechanism, virus-induced changes in ERGIC-53 trafficking were identified, resulting in an ERGIC-53 appearance at the plasma membrane where the budding arenavirus incorporated the lectin for an improvement of infectivity [39].

Such alterations in ERGIC-53 trafficking could not be detected in HBV-replicating cells unless the ER export was blocked. Moreover, we were unable to verify a colocalization between ERGIC-53 and the viral envelope under natural settings, which was unanticipated given its pro-viral role in HBV propagation. ERGIC-53 is known to capture its cargoes in a pH- and Ca^2+^-dependent manner at the ER and releases them in the ERGIC, where the lower pH and Ca^2+^ concentrations diminish its affinity for glycoproteins [50,53]. Hence, within the ERGIC, the HBV envelope may rapidly dissociate from ERGIC-53, thereby rendering their colocalization below the limit of detection. Similarly, we failed to detect HBV envelope structures within Golgi stacks, although the N-glycan pattern of the extracellular virions implicate a transit through this compartment. One likely interpretation may be that HBV assembly and budding are inextricably linked and rate-limiting processes. Once disseminated from the host membrane, HBV viral particles may be rapidly exported out of the cell.

There is concordant evidence that HBV budding takes place at intracellular membranes rather than at the plasma membrane. In regard to our microscopy studies, the viral envelope accumulated in crescent-shaped ER-associated structures where it actively recruited the viral core/capsid. This CS structure shares hallmarks of viral interaction platforms, as an envelopment-defective core mutant (C.K96A) failed to be attracted. Our results are in accord with recent confocal microscopy and CO-IP studies that demonstrated a lack of interaction between the HBV L protein and the C.K96A mutant [64]. Nonetheless, the C.K96A mutant has been shown to warrant the formation and secretion of genome-free empty virions, another HBV particle type with relatively unknown functions [65]. Unlike complete virions, empty virion assembly requires only the S envelope protein and may thus proceed via distinct mechanisms and pathways. Our observation that the characteristic recruitment of the viral core to the L-specific CS structure and even the formation of enveloped HBV-replicating nucleocapsids preserved upon the ER exit block led us to infer that HBV assembly initiates at the ER. Therefore, ERGIC-53 and the COPII coat proteins presumably come into play. However, their temporal and spatial modes of action are yet ambiguous. In particular, we still do not know whether Sec24A may act as a direct COPII adaptor for the viral envelope or as an indirect adaptor recognizing the envelope-associated ERGIC-53 cargo receptor. A bipartite use of adaptors and receptors has been suggested to amplify the cargo capture into vesicles [66].

In regard to its ESCRT dependency and cell-imaging studies, HBV has been viewed as a MVB-budding virus [20,21,22,23,24,67]. However, the tracing of viral proteins or particles within the MVB/endosomal system does not mandatorily ascertain their use as viral exit portals. Rather, HBV progeny particles and proteins have been shown to be destructed within the endo-lysosomal system [68,69,70]. Many viruses have been shown to hijack and recruit the ESCRT machinery to wrap membranes around new virus particles at their particular budding sites [25]. Consistent with this, we found that Vps4A, the terminal ATPase of the ESCRT machinery, was specifically recruited by the HBV-specific ER-associated CS structure. In addition, the functional inactivation of the ESCRT-II complex inhibited the HBV release prior to N-glycan processing occurring within the Golgi complex. While the precise itinerary of HBV out of the cell remains to be decoded, our results altogether implicate that ESCRT-driven HBV assembly and budding may occur at the ER.

This study is the first to document an essential role of the cellular ERGIC-53 lectin in HBV’s pathogenic life cycle. ERGIC-53 acts in teamwork with selective components of the COPII machinery, encompassing Sec24A, Sec23B and Sar1, to drive HBV ER export. Viral SVP secretion bypasses the need of this trafficking factor, implicating that ERGIC-53 resembles a checkpoint for the spatiotemporal export pathway segregation of HBV viral and spherical subviral particles. The characterization of the HBV-ERGIC-53 interaction identifies a pathogen-derived N-glycan as a crucial but not exclusive ligand of the lectin. The fact that the loss of ERGIC-53 or its function is well-tolerated in humans renders this protein as an attractive cellular target for therapeutic antiviral intervention.

## Figures and Tables

**Figure 1 cells-09-01889-f001:**
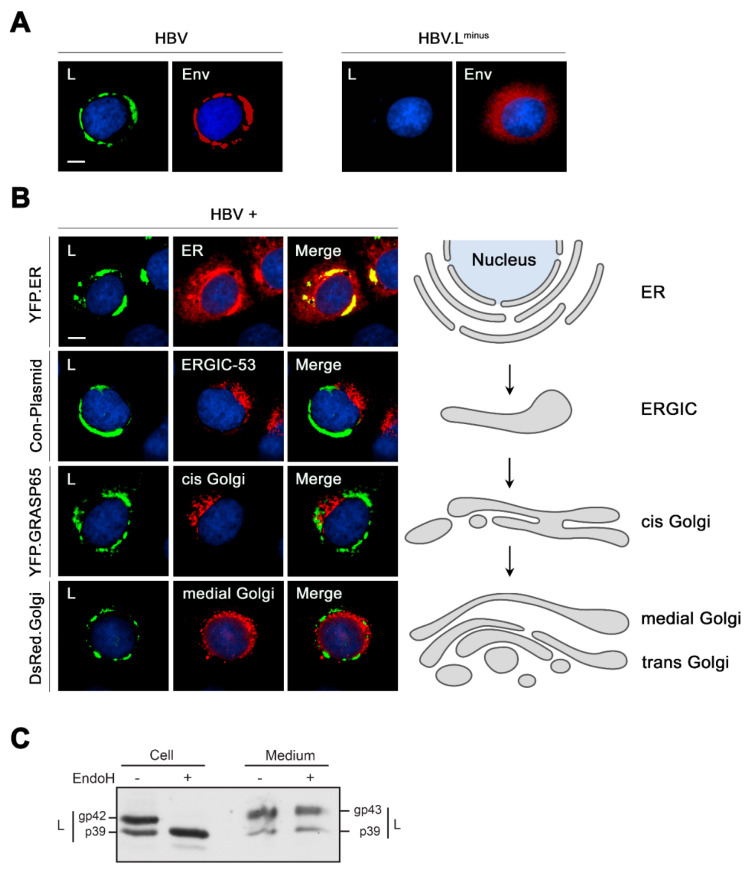
Trafficking analysis of hepatitis B virus (HBV) L through the cell secretory system. (**A**) HuH-7 cells were transfected with the wild-type (wt) HBV replicon construct (HBV) or the mutant replicon HBV.L^minus^ defective in L protein synthesis. Three days post-transfection, cells were fixed with paraformaldehyde (PFA), permeabilized and reacted with L-specific antibodies (57761, L, green) and an S domain-specific antibody, recognizing all three L, M and S proteins (K38, Env, red). Bar, 10 µM. (**B**) HuH-7 cells were cotransfected with the plasmid pHBV* replicon together with control DNA or plasmids encoding YFP.ER, green fluorescent protein (GFP).GRASP65 or DsRed.Golgi, as indicated. PFA-fixed cells were reacted with antibodies against the preS1 domain of L (K1350, green). To visualize endoplasmic reticulum-Golgi intermediate compartment (ERGIC) structures, cells were co-stained with an ERGIC-53-specific antibody (red). The autofluorescence pattern of YFP.ER, GFP.GRASP65 and DsRed.Golgi are shown in (pseudo-colored) red, and DNA staining is depicted in blue. Overlaid staining patterns are displayed in the Merge panels. Bar, 10 µM. The cartoon on the right depicts the anterograde organization of the cell secretory pathway. (**C**) Lysates (Cell) and concentrated supernatants (Medium) of HBV-replicating HuH-7 cells were mock-treated (−) or treated (+) with Endoglycosidase H (EndoH) to inspect the N-glycan pattern of L. Samples were examined by Western blotting (WB) using L-specific (K1350) antibodies. The nonglycosylated (p39) and glycosylated (gp42/gp43) forms of L are indicated.

**Figure 2 cells-09-01889-f002:**
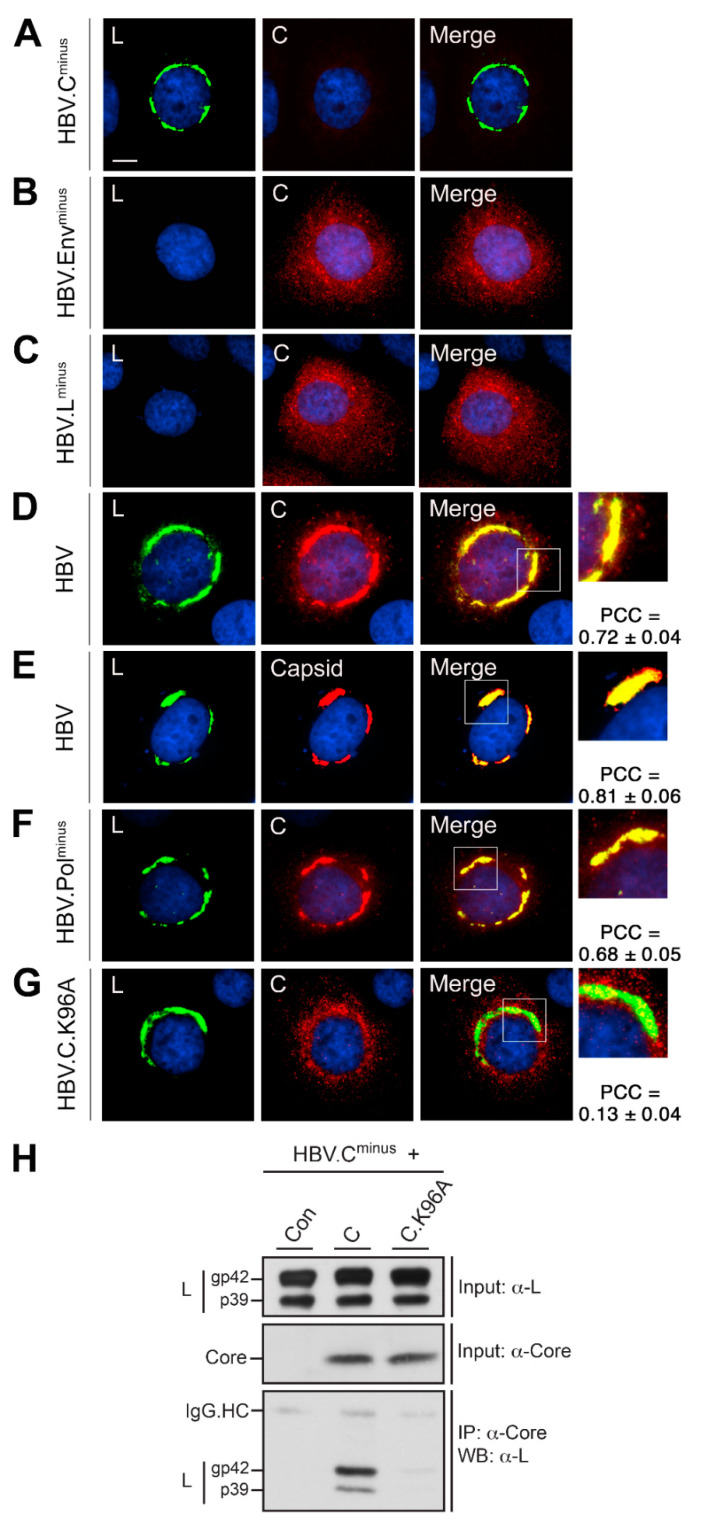
HBV L recruits envelopment-competent core/capsids to ER-associated structures. (**A**) HuH-7 cells were transfected with the wild-type (wt) HBV replicon construct (HBV) or mutant replicons defective in core protein expression (HBV.C^minus^), envelope protein expression (HBV.Env^minus^), L envelope protein expression (HBV.L^minus^) or polymerase activity (HBV.Pol^minus^). Cells shown in panel F were cotransfected with the core-defective replicon and a plasmid encoding the core mutant C.K96A. PFA-fixed cells were double-stained with mouse antibodies against L (57761, L, green) and rabbit antibodies against core (B0586, C, red), followed by staining with Alexa Fluor 488-conjugated anti-mouse and Alexa Fluor 546-conjugated anti-rabbit antibodies (**A**–**G**). In the panels shown in **E**, cells were reacted with rabbit anti-L antibodies (K1350, L, green) and mouse antibodies recognizing the viral capsid rather than the core protein (3HB17, Capsid, red). Overlaid images are displayed in the Merge panels, and nuclei staining is shown in blue. Bar, 10 µM. Outlined areas are shown at larger magnifications. For quantitative colocalization analysis (QCA), Pearson´s correlation coefficient (PCC) values were calculated and represent the mean ± SD. (**H**) HuH-7 cells were transfected with HBV.C^minus^ together with control plasmids (Con), the wt core (C) or mutant C.K96A genes. Cell extracts were prepared by CHAPS-lysis and assayed by L- and core-specific WB to monitor protein expression (Input). For immunoprecipitation (IP), lysates were incubated with anti-core antibodies (K45), followed by immunoblotting with the L-specific MA18/7 antibody. IgG.HC denotes IgG heavy chains. Coimmunoprecipitation (CO-IP) analyses were done in triplicate, and a representative blot is shown.

**Figure 3 cells-09-01889-f003:**
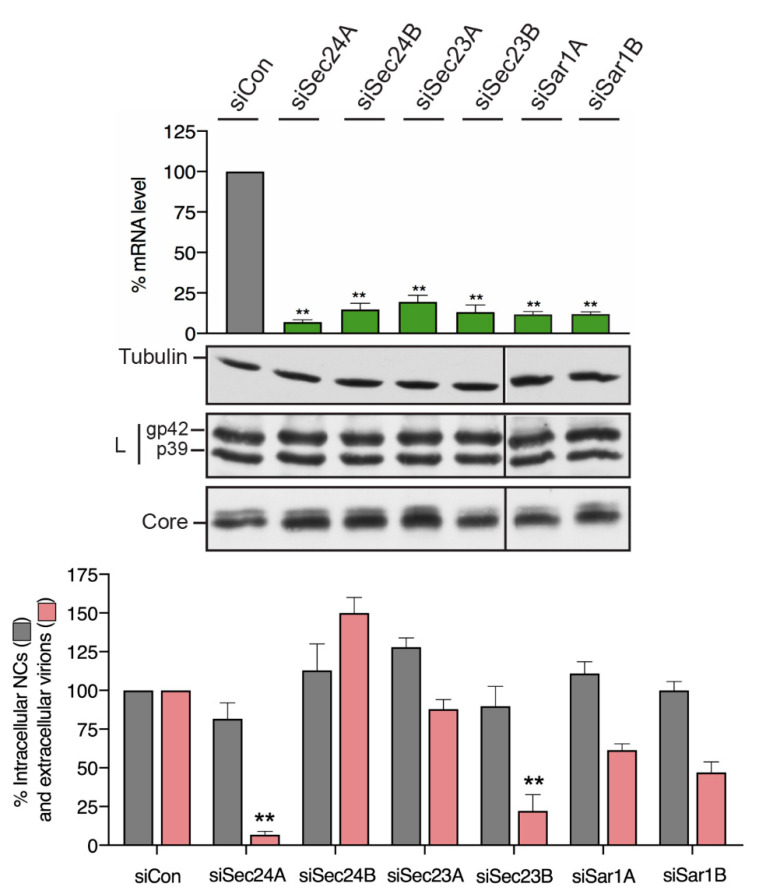
HBV release requires COPII components. HuH-7 cells were treated with control siRNA (siCon) or siRNAs targeting Sec24A, Sec24B, Sec23A, Sec23B, Sar1A or Sar1B for 48 h and retransfected with the pHBV* replicon. After an additional 72 h, lysates and supernatants were harvested. To check for depletion, transcript levels of the RNA interference (RNAi)-targeted genes were measured by qRT-PCR and demonstrated in percent amount relative to control cells (*n* = 4, ± SD). To monitor protein synthesis, cell lysates were examined by WB using tubulin-, L- (K1350) and core-specific (K46) antibodies. The nonglycosylated (p39) and glycosylated (gp42) forms of L are indicated. HBV release was measured by envelope-specific IP of supernatants using anti-L (K1350) and anti-S (K38) antibodies and real-time multiplex PCR measurements of the viral genomes (extracellular virions). Nonenveloped intracellular nucleocapsids (intracellular NCs) were immunoprecipitated with anti-capsid (K45) antibodies and analyzed by PCR. PCR results were demonstrated in the percent amount relative to siCon-transfected cells. Error bars indicate the standard deviations from the mean of three experiments measured in duplicates. ** *p* < 0.01 compared to control.

**Figure 4 cells-09-01889-f004:**
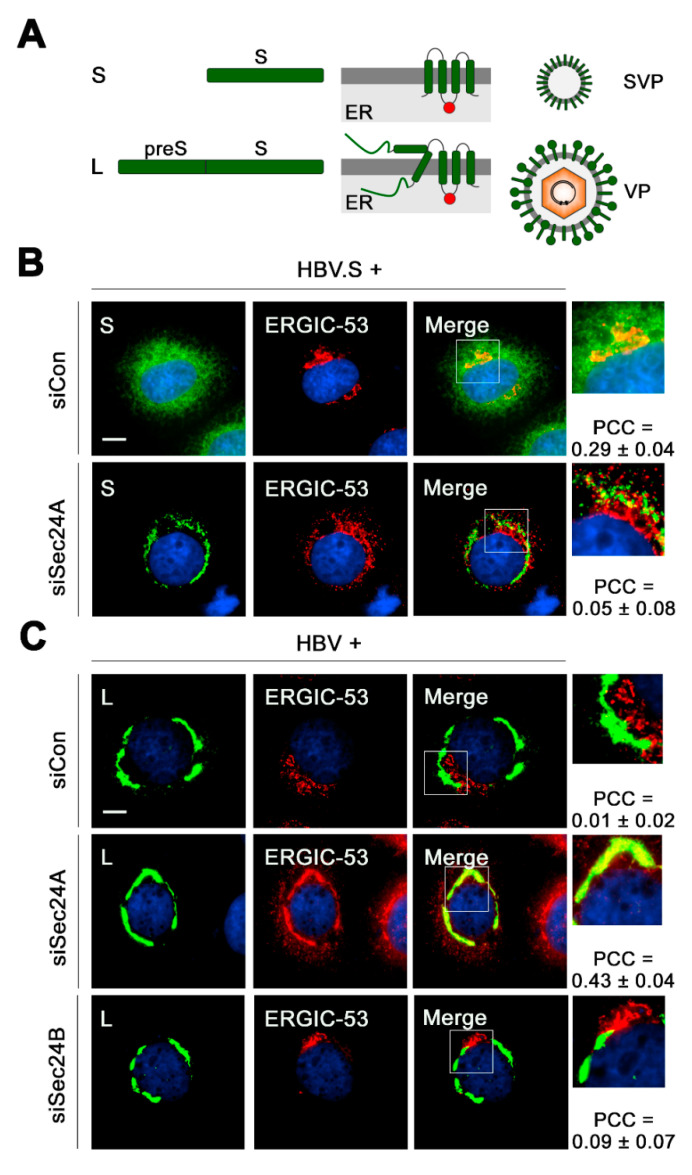
Viral HBV L, but not subviral HBV S, colocalizes with ERGIC-53 upon ER export block. (**A**) Domain structures of the S and L proteins (left), their transmembrane topology at the ER (middle) and HBV subviral particle (SVP) and VP types (right). The red circles depict the N-glycan acceptor N146. (**B**,**C**) Cells were treated with the indicated RNA duplexes for 2 days and retransfected with pHBV.S (**B**) or pHBV* (**C**). After an additional two days, cells were fixed with PFA and stained with either rabbit anti-S (K38, S, green) (**B**) or rabbit anti-L (57761, L, green) (**C**) plus mouse anti-ERGIC-53 (red) antibodies. DNA staining is shown in blue. Overlaid images (yellow) are displayed in the Merge panels with magnifications and PCC data. Bar, 10 µM.

**Figure 5 cells-09-01889-f005:**
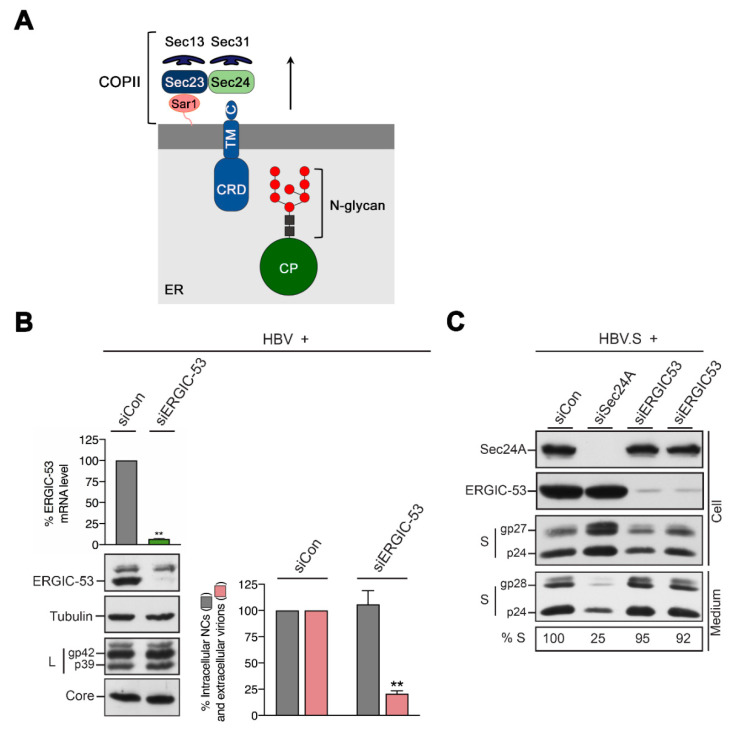
ERGIC-53 controls the egress of HBV viral particles but not of HBV spherical subviral particles. (**A**) Model depicting the structure of ERGIC-53 in association with cargos and COPII. ERGIC-53 is a type I integral membrane protein with a C-terminal transmembrane domain (TM) and a luminal cargo recognition domain (CRD) interacting with N-linked high-mannose glycans of cargo proteins (CP) within the ER. Its cytoplasmic ER exit motif (**C**) binds to Sec24, the cargo adaptor of COPII. GTP activation of Sar1 leads to the formation of the Sec23/24 heterodimer, cargo capture and coat polymerization by the Sec13/31 complex. (**B**) The treatment of HuH-7 cells with the control (siCon) or ERGIC-53-specific siRNAs for 2 days and retransfection with the HBV* replicon construct for an additional 3 days. The degree of ERGIC-53 depletion was measured by qRT-PCR of the transcript levels and WB analysis. Cellular lysates and supernatants were subjected to tubulin-, L- and core-specific WB and to the virion production assay (*n* = 3, ± SD). (**C**) Treatment of HuH-7 cells with control (siCon), Sec24A- or ERGIC-53-specific siRNA duplexes for 3 days and retransfection with an haemagglutinin (HA)-tagged S construct (HBV.S) for an additional 2 days. The ERGIC-53-specific RNAi experiments were done in duplicate. To monitor depletion, cell lysates were examined by Sec24A- and ERGIC-53-specific WB. Cell lysates (Cell) and concentrated cell supernatants (Medium) were probed by anti-HA immunoblotting. The nonglycosylated (p24) and glycosylated (gp27 and gp28) forms of HBV.S are depicted.

**Figure 6 cells-09-01889-f006:**
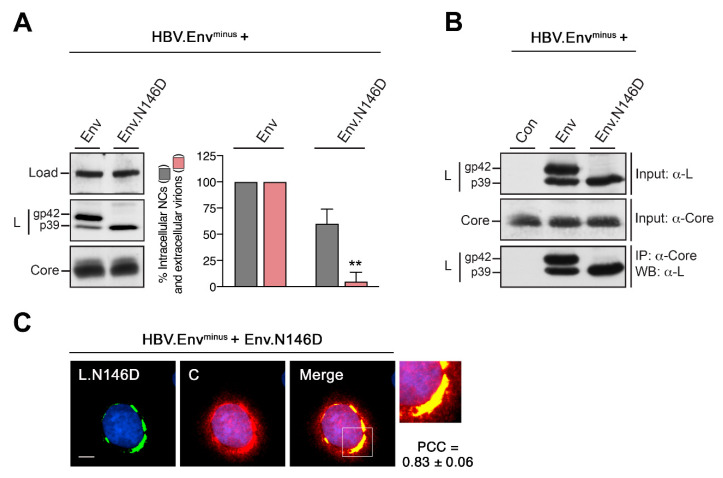
Abrogation of N146-linked glycosylation inhibits HBV release without affecting L/core interaction. (**A**) HuH-7 cells were cotransfected with the envelope-defective HBV replicon (HBV.Env^minus^) complemented with expression constructs encoding the wt envelope (Env) or mutant Env.N146D proteins. Three days post-transfection, cellular lysates and supernatants were subjected to L- and core-specific WB analysis and to the virion production assay (*n* = 3, ± SD). (**B**) HuH-7 cells were transfected with HBV.Env^minus^ together with control plasmids (Con), the wt envelope (Env) or mutant Env.N146D constructs. To verify equal protein synthesis, lysates were examined with anti-L (K1350) and anti-core (K46) antibodies. For CO-IP analysis, lysates prepared with CHAPS were reacted with anti-core antibodies (K45) followed by L-specific (MA18/7) WB. (**C**) HuH-7 cells, cotransfected with HBV.Env^minus^ and the mutant Env.N146D expression construct, were processed for immunofluorescence (IF) and stained with anti-L (57761, L, green) and anti-core (B0586, C, red) antibodies. An overlaid image is displayed in the Merge panel, with an outlined area shown at larger magnification. For QCA, PCC values were calculated and represent the mean ± SD. Bar, 10 µM.

**Figure 7 cells-09-01889-f007:**
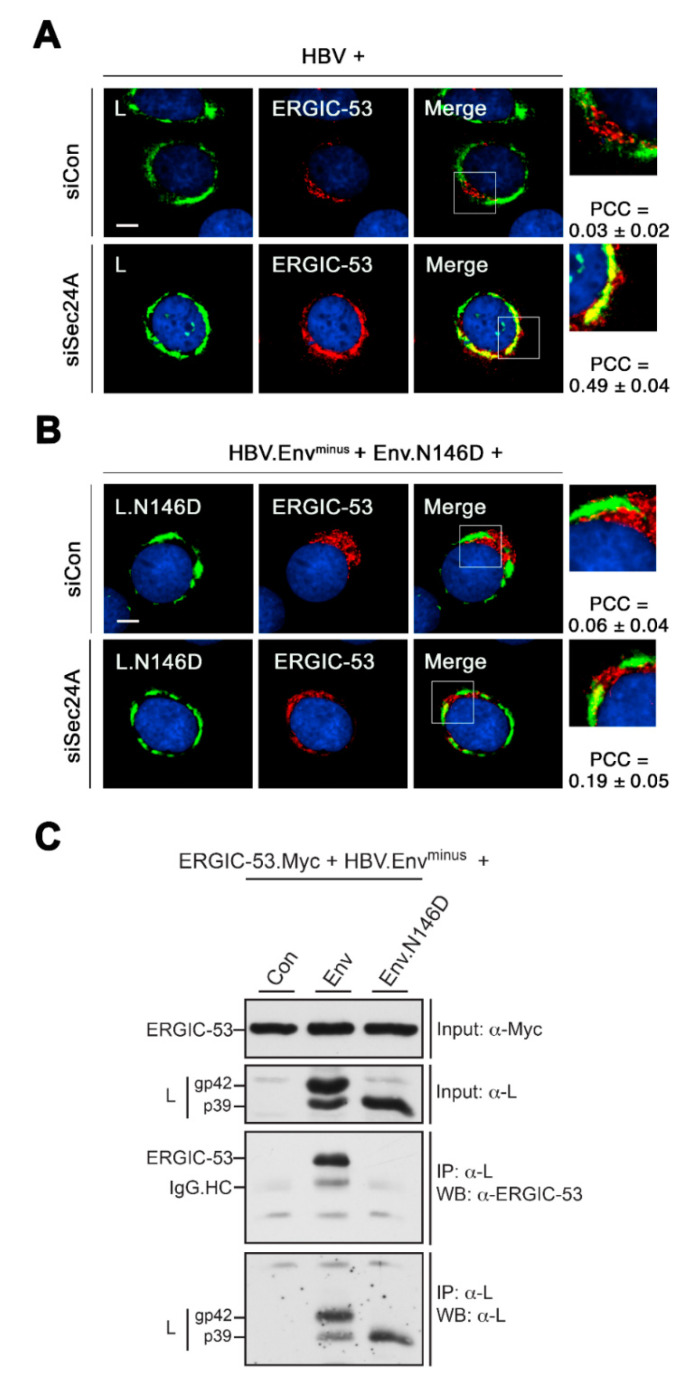
HBV L colocalizes and interacts with ERGIC-53 in a N146-glycan-dependent manner. (**A**,**B**) IF analyses of HuH-7 cells treated with control or siSec24A-specific duplexes for two days followed by transfection with the wt HBV replicon (**A**) or glycan-defective HBV replicon (**B**) for an additional 2 days. PFA-fixed cells were stained with anti-L (K1350, L, green) and anti-ERGIC53 (red) antibodies. Merged images are shown in the right panels, with outlined areas shown at larger magnification. For QCA, PCC values were calculated and represent the mean ± SD. Bar, 10 µM. (**C**) For CO-IP analysis, HuH-7 cells were cotransfected with a Myc-tagged version of ERGIC-53 (ERGIC-53.Myc) and HBV.Env^minus^ complemented with either the wt (Env) or mutant Env.N146D constructs. To test for equal input amounts, cell extracts prepared by Triton lysis were probed by WB with anti-Myc and anti-L (K1350) antibodies. Lysates were precipitated with mouse anti-L (57761) antibodies and analyzed by WB using rabbit anti-ERGIC-53 (E1031) antibodies. To confirm equal IP efficiencies, the same blot was reprobed with anti-L (K1350).

**Figure 8 cells-09-01889-f008:**
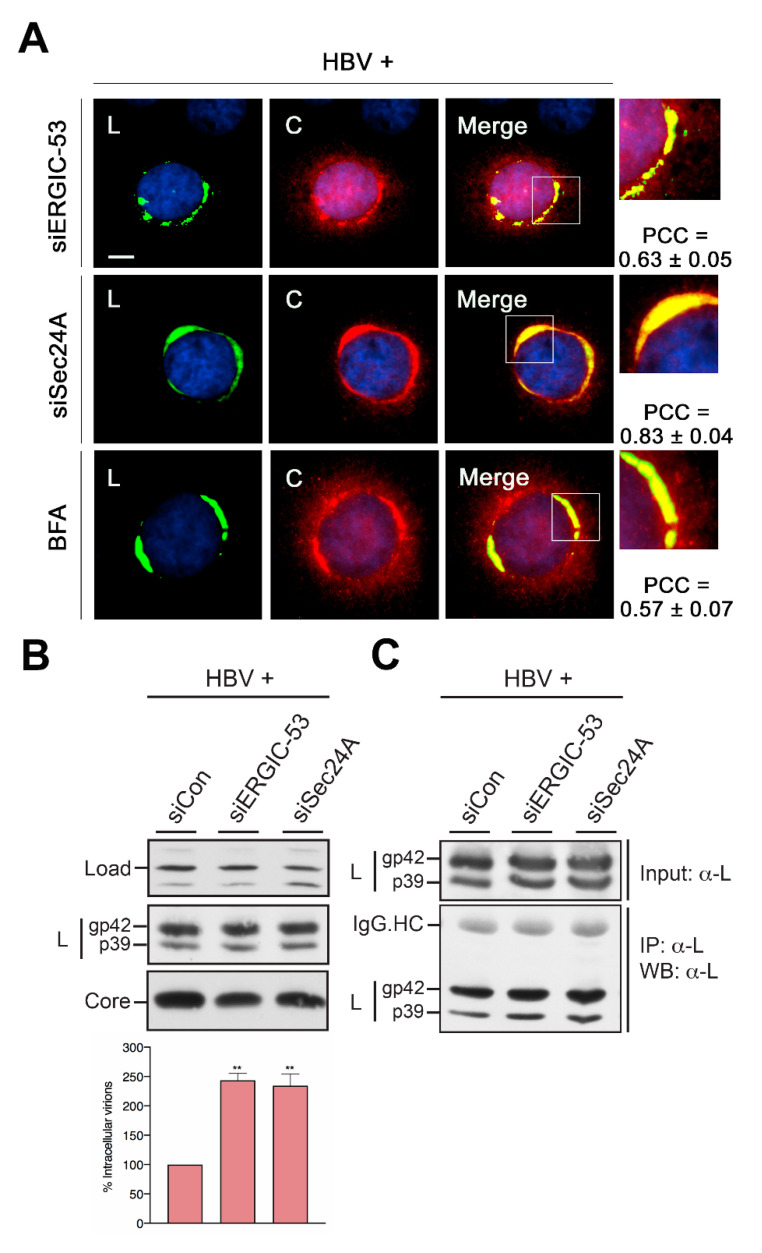
HBV envelopment of nucleocapsids occurs in ERGIC-53- and Sec24A-depleted cells. (**A**) ERGIC-53 KD, Sec24A KD or BFA-treated cells were transfected with the HBV replicon, processed for IF and stained with L- (57761, L, green) and core-specific (B0586, C, red) antibodies. IF analyses and quantifications were done as above. (**B**) HuH-7 cells were treated with siCon-, siERGIC-53- or siSec24A-specific duplexes, followed by transfection with pHBV*. Cells were lysed by repetitive freeze-thaw cycles, and intracellular enveloped virions were precipitated with L-specific antibodies in the absence of detergents and assayed by PCR. Error bars indicate the standard deviations from the mean of two experiments measured in duplicates. To monitor HBV protein expression, cell extracts were probed by L- and core-specific WB, as above. A nonspecific band stained by the antisera served as a control for identical gel loading. (**C**) To probe for IP efficiencies, cells were treated and processed exactly as in **B**, and immunoprecipitated samples were analyzed by L-specific WB (MA18/7). The p39 and gp42 forms of L are indicated. Experiments were done in duplicate, and representative blots are shown. ** *p* < 0.01 compared to control.

**Figure 9 cells-09-01889-f009:**
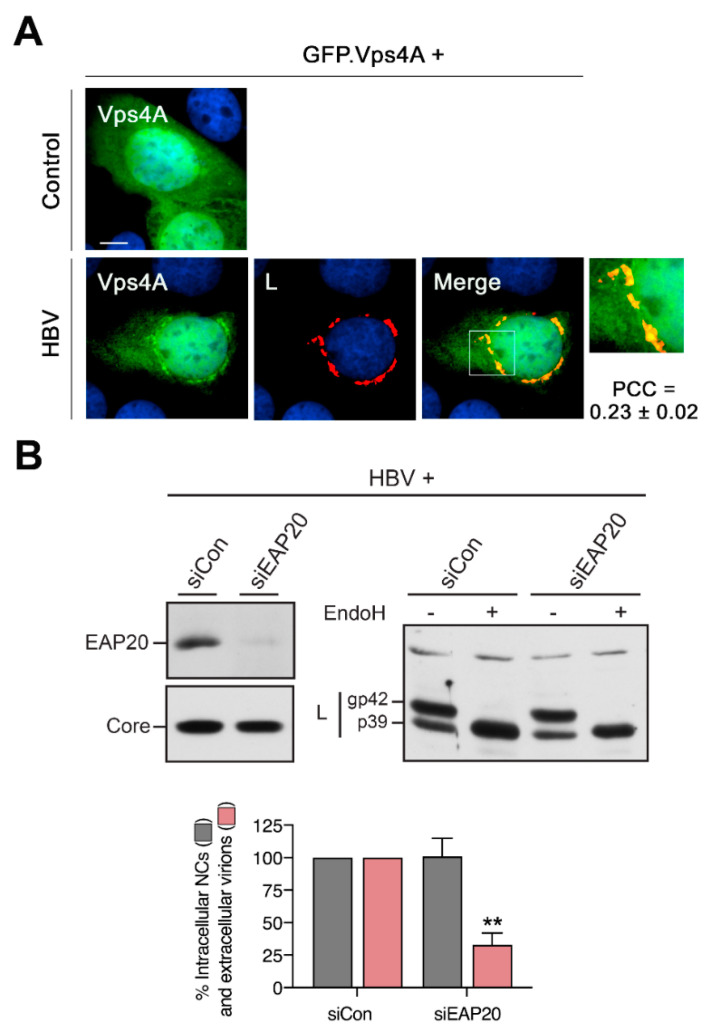
Recruitment of ESCRT by the HBV-specific CS structure of L. (**A**) Cells were transfected with a GFP-tagged version of the Vps4A ATPase together with the control plasmid or the HBV replicon, followed by PFA fixation. The autofluorescence pattern of GFP.Vps4A is shown in green, anti-L labeling (57761, L) is in red and PCC values are indicated. (**B**) For RNAi of the ESCRT-II complex, cells were treated with EAP20-specific duplexes, transfected with the HBV* replicon and subjected to the virion production assay (*n* = 3, ± SD) exactly as described above. For a reference, a siCon treatment of cells was included. Lysates were probed by anti-EAP20 and anti-core (K46) immunoblotting. The same lysates were mock-treated or treated with EndoH prior to anti-L (K1350) WB. The L-specific p39 and gp42 forms are indicated. ** *p* < 0.01 compared to control.

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
