# Peer review of "Hepatitis B Virus Exploits ERGIC-53 in Conjunction with COPII to Exit Cells"

_cells, 2020, doi:10.3390/cells9081889_

Round 1
Reviewer 1 Report
In this study, Zeyen et al. used the imaging method and siRNA knocking down to study the mechanisms of HBV assembly and trafficking. They found that HBV takes advantage of ER export machinery COPII for viral trafficking and egress. They also characterized that ERGIC-53 is critical for HBV virion egress via L protein interaction in an N-glycosylation-dependent manner. However, in this study, some controls were inappropriately used, which might mislead the conclusions. Other comments are listed below.
Major points:
- In the HBV field, SVP means subviral particles, including both spherical (formed by S alone), and filamentous (formed by L and S mostly) HBsAg particles. In this paper, the author used SVP referred to as the spherical particles formed by S-HBs alone and the envelope protein referred to L-HBs. To make it clearer, the authors need to clarify that spherical SVPs formed by S-HBs and L-HBs protein instead of SVP and envelope protein, respectively, or other expressions to avoid any confusion.
- In Fig.1, the author used the HBV full-length replicon to track the L-HBs in the secretory pathway. The full-length replicon expresses L, M, and S. The majority of this signal of L shown in Fig.1A will form the filament SVPs, and some of L forms virions. It is confusing that in Lines 284-286 said that “Worth mentioning, a similar pattern emerged when S domain-specific antibodies were used for staining (data not shown), indicating that the CS structure is specific for the viral envelope rather than for L.”. As shown in the Zeyen et al., (2020, Cellular Microbiology), HA-S does not form the crescent-shaped structure. Using the S-HBs-specific antibody will detect the L, M, and S, so it is not surprising that both antibodies have similar patterns. The L-antibody used in this study presumably is specific to L-HBs. Thus, the interpretation for the Fig.1A is the crescent-shaped pattern is specific for L. To interpret the result more concisely, the authors should include HBV L(minus) as the negative control in Fig.1A. The viral envelope here seems referred to L, M, and S. As in the 1st comments, the authors need to correct their terms in viral envelope and SVPs to make it clearer.
- 2 (A) and (B) are important negative controls. Although no signals of HBc (or core) in Fig.2A and no L signal in Fig.2B, the authors should include panels of L, C, and merged pictures, as shown in Fig.2C-2F. The authors claimed L-recruits core/capsid to the crescent-shaped ER-associated compartment. However, they used HBV.Envminus, which lacks expression of L, M, and S. Thus, in this experiment, the results might be over-explained, the authors need to include HBV-Lminus as the negative control, instead of using HBV.Env(minus).
- In Fig.2(E), HBV.Polminus was found to have HBc co-localized with L at the crescent-structure. As reported by Ning et al., (2011, PLoS Pathogens), HBV.Polminus does not secrete DNA-containing virions due to defective HBV pgRNA packaging; however, the HBV.Polminus does support the HBV genome-free virions (empty virions) secretion. Ning et al., (2018, J. Virol) reported the same phenotype in HBc-K96A mutant, which secrete empty virions but not DNA-containing virions. Here, the authors showed the different results of HBV.Polminus and HBV.Cminus co-transfected with HBc-K96A, in terms of the capsid and L-HBs co-localization in the crescent-structure. The signal authors detected here in HBV.Polminus is contributed by naked capsids (formed by HBc alone) but not the nucleocapsid (devoid of viral genome). The results could be explained by the complexities of how HBV viral envelop and capsid interacts. Thus, the crescent structure is likely to mirror the site of the capsid (naked capsid or nucleocapsid) and L-HBs interaction, instead of HBV assembly sites as said in Line 349, which might be over-interpreted. The authors could discuss this in the Discussion.
- In section 3.8, the authors used L-specific IP in the absence of detergents and then performed PCR to quantify the intracellular HBV virions. The method raises a concern regarding the IP efficiency since L-protein is part of filament SVPs, which are more abundant than the virions. L-specific IP will pull-down more filament SVPs than HBV virions. The authors showed c.a. 2.5-fold difference in intracellular virions between siCon and siERGIC53 or siSec24A groups by PCR, this c.a. 2.5-fold difference can be explained by pull-down efficiency. Whether knock-down of ERGIC53 or Sec24A affects filament SVPs is not shown in the current study. Viral envelope and SVPs are difficult to distinguish by IP. To address the effect ERGIC53 and Sec24A on filament SVP secretion, the authors need to include the construct expressing LMS (the construct from C. Sureau) as the control for IP and include the immunoblot results to show the pull-down efficiency. The panel 8B(left) only shows intracellular levels (i.e., input of IP), which is not helpful in addressing the IP efficiency. Thus, adding the experiments for demonstrating IP-efficiency will strengthen that ERGIC53 and Sec24A affect intracellular viral transport.
Minor points:
- Line 12: “characterization of intersections…” should be “characterization of interactions…”.
- Line 30-31: “About 240 million people are persistently infected, …” should use the most recent data from WHO, which is about 257 million.
- Line 45-46: “This replication intermediate then serves as a template for RNA transcription and translation into the viral proteins in the cytoplasm.” can be rephrased into “This replication intermediate serves as the template for viral RNA transcription.”
- Line 84: “The viral S envelope protein alone is sufficient for the production of SVPs that are formed by…” should precisely use spherical SVPs.
- Line 87: “spherical SVPs does neither require N146-linked N-glycans” should be “spherical SVPs does neither require N146-linked glycans”.
- Line 106: “…recognizes the viral envelope in a N146-glycan-dependent manner…” should be “…recognizes the viral envelope in an N146-linked glycan-dependent manner”.
- Line 274: “Unlike those of HBV subviral envelope particles (SVPs), intracellular trafficking pathways of the viral envelope are poorly understood.” This sentence is unclear and could be rephrased into “Unlike those of HBV spherical subviral envelope particles (SVPs) formed by S alone, intracellular trafficking pathways of the L is poorly understood.” See the major comments above.
- Line 363: “(B) HuH-7 cells were transfected with HBV.Cminus… “ should be “(G) HuH-7 cells were transfected with HBV.Cminus…”.
- Line 384: delete the nucleocapsids and use NCs.
- Line 420: “is transported out of the ER to the ERGIC in control cells (Figure 4A).” should be “is transported out of the ER to the ERGIC in control cells (Figure 4B)”.
- Line 422: “…that no longer coincided with ERGIC-53 signals (Figure 4A).” should be “that no longer coincided with ERGIC-53 signals (Figure 4B).”
- Line 440-441: “In contrast, the HBV viral envelope could not be traced within the ERGIC as no colocalization between L and ERGIC-53 was detectable in control cells (Figure 4B).” should be “In contrast, the HBV Large envelope (or L) protein could not be traced within the ERGIC as no colocalization between L and ERGIC-53 was detectable in control cells (Figure 4C).”.
- In Line532, authors wrote transfected with wt or Env.N146D replicons, but in Fig.7B labeled as the HBV.Envminus + Env.N146D, which means co-transfection. Please correct it properly.
- Line 717-720: “There, ERGIC-53 and the COPII coat proteins presumably come into play, but their temporal and spatial mode of action is yet ambiguous. In particular, we still do not know whether Sec24A may act as a direct COPII adaptor for the viral envelope or as an indirect adaptor, recognizing the envelope-associated ERGIC-53 cargo receptor.” should be “Therefore, ERGIC-53 and the COPII coat proteins presumably come into play. However, their temporal and spatial mode of action is yet ambiguous. In particular, we still do not know whether Sec24A acts as a direct COPII adaptor for the viral envelope or as an indirect adaptor, recognizing the envelope-associated ERGIC-53 cargo receptor.”
Author Response
Reviewer #1: Major points
- In the HBV field, SVP means subviral particles, including both spherical (formed by S alone), and filamentous (formed by L and S mostly) HBsAg particles. In this paper, the author used SVP referred to as the spherical particles formed by S-HBs alone and the envelope protein referred to L-HBs. To make it clearer, the authors need to clarify that spherical SVPs formed by S-HBs and L-HBs protein instead of SVP and envelope protein, respectively, or other expressions to avoid any confusion.
Yes, we thank the reviewer and apologize for any confusion. For clarification, we now used the term “HBV spherical subviral particles, spherical SVPs or HBV spheres” throughout the manuscript. For further differentiation between viral and subviral particles, we stated: “HuH-7 cells were transiently transfected with an expression construct encoding only the S envelope gene (HBV.S). Notably, this is an established approach to study exclusively the fate of subviral HBV spheres” (lines 431-433) and “In contrast, the HBV L protein, synthesized in HBV-replicating cells, could not…..”(line 455). In addition, we now briefly introduced HBV filaments in the introduction (lines 92-95).
- In Fig.1, the author used the HBV full-length replicon to track the L-HBs in the secretory pathway. The full-length replicon expresses L, M, and S. The majority of this signal of L shown in Fig.1A will form the filament SVPs, and some of L forms virions. It is confusing that in Lines 284-286 said that “Worth mentioning, a similar pattern emerged when S domain-specific antibodies were used for staining (data not shown), indicating that the CS structure is specific for the viral envelope rather than for L.”. As shown in the Zeyen et al., (2020, Cellular Microbiology), HA-S does not form the crescent-shaped structure. Using the S-HBs-specific antibody will detect the L, M, and S, so it is not surprising that both antibodies have similar patterns. The L-antibody used in this study presumably is specific to L-HBs. Thus, the interpretation for the Fig.1A is the crescent-shaped pattern is specific for L. To interpret the result more concisely, the authors should include HBV L(minus) as the negative control in Fig.1A. The viral envelope here seems referred to L, M, and S. As in the 1stcomments, the authors need to correct their terms in viral envelope and SVPs to make it clearer.
We appreciate this helpful comment. As requested we now have included an HBV.Lminus replicon construct as negative control in the IF studies. The results are now shown in a modified Figure 1 (new Figure 1A) and described in the text (lines 295-300). Yes, upon ablation of L protein expression, the CS structure got lost, indicating that this structure is specific for L (line 300). We therefore have deleted the sentence: “Worth mentioning, a similar pattern emerged when S domain-specific antibodies were used for staining (data not shown), indicating that the CS structure is specific for the viral envelope rather than for L.” Yes, we agree that this structure may present both, HBV filaments and virions. However, given that this structure recruits core/capsids – that are only needed for virion but not filament formation – we feel that this structure likely resembles “virus production niches”.
- 2 (A) and (B) are important negative controls. Although no signals of HBc (or core) in Fig.2A and no L signal in Fig.2B, the authors should include panels of L, C, and merged pictures, as shown in Fig.2C-2F. The authors claimed L-recruits core/capsid to the crescent-shaped ER-associated compartment. However, they used HBV.Envminus, which lacks expression of L, M, and S. Thus, in this experiment, the results might be over-explained, the authors need to include HBV-Lminusas the negative control, instead of using HBV.Env(minus).
As suggested, we now included the corresponding core-, L- and merge-specific staining pattern (modified Figures 2A and 2B). In addition, we also included imaging data of the HBV.Lminus construct, as requested (new Figure 2C). All together, these data confirm our original interpretation that L recruits core to the CS structure where capsid/envelope interactions take place.
- In Fig.2(E), HBV.Polminuswas found to have HBc co-localized with L at the crescent-structure. As reported by Ning et al., (2011, PLoS Pathogens), HBV.Polminus does not secrete DNA-containing virions due to defective HBV pgRNA packaging; however, the HBV.Polminus does support the HBV genome-free virions (empty virions) secretion. Ning et al., (2018, J. Virol) reported the same phenotype in HBc-K96A mutant, which secrete empty virions but not DNA-containing virions. Here, the authors showed the different results of HBV.Polminus and HBV.Cminus co-transfected with HBc-K96A, in terms of the capsid and L-HBs co-localization in the crescent-structure. The signal authors detected here in HBV.Polminus is contributed by naked capsids (formed by HBc alone) but not the nucleocapsid (devoid of viral genome). The results could be explained by the complexities of how HBV viral envelop and capsid interacts. Thus, the crescent structure is likely to mirror the site of the capsid (naked capsid or nucleocapsid) and L-HBs interaction, instead of HBV assembly sites as said in Line 349, which might be over-interpreted. The authors could discuss this in the Discussion.
We would like to reply to this comment in subunits.
(A) The HBV.Polminus replicon construct. As we have already stated in our original manuscript, this mutant carrying a D540H substitution “had been shown to be defective in reverse transcriptase activity” (original manuscript, lines 118-120, 331). We would like to stress that this mutant is different from the HBV.Polminus mutant, studied by Ning et al. (2011, PLoS Pathogens) that does not express a polymerase protein due to frameshift mutation after codon 30 in the core gene. Accordingly, the two mutant constructs appear to differ in their “genome content”, i.e., no genome (the mutant used by Ning et al.) or a pgRNA genome (the mutant used herein) which may affect their interaction capacities with the viral L protein. For clarification, we have now emphasized in the revised manuscript that our mutant is defective in replication but competent in pgRNA packaging (i.e., not devoid of a viral genome) (lines 354-355) and added a new reference # 43. Accordingly, our results show that pgRNA-containing capsids can be recruited to the L-specific CS structure.
(B) Yes, the recent identification of genome-free, empty HBV virions is an exciting hot topic. Since capsids generated in our HBV.Polminus replicon system are not genome-free, they are unlikely to have access to the empty virion assembly and secretion pathway. As suggested, we have now briefly introduced genome-free virions in the revised discussion (lines 738-741) and added a new reference # 65 (Ning et al., 2018, J. Virol.). With regard to this new reference, we now discuss that the structural requirements for the formation and release of genome-free virions and complete virions are clearly different, and hence may resemble different assembly pathways (lines 740-741).
(C) Concerning our results obtained with “HBV.Cminus co-transfected with HBc-K96A”, we now also added a new reference # 64 (Pastor et al., 2019). This study reported results that are entirely concordant with our observations, i.e., that the HBc-K96A mutant failed to interact and colocalize with the L protein in a cellular context.
(D) Yes, we agree that “the crescent structure is likely to mirror the site of the capsid/L interaction, instead of HBV assembly sites” and substituted the term “assembly sites” by interaction sites (line 366).
- In section 3.8, the authors used L-specific IP in the absence of detergents and then performed PCR to quantify the intracellular HBV virions. The method raises a concern regarding the IP efficiency since L-protein is part of filament SVPs, which are more abundant than the virions. L-specific IP will pull-down more filament SVPs than HBV virions. The authors showed c.a. 2.5-fold difference in intracellular virions between siCon and siERGIC53 or siSec24A groups by PCR, this c.a. 2.5-fold difference can be explained by pull-down efficiency. Whether knock-down of ERGIC53 or Sec24A affects filament SVPs is not shown in the current study. Viral envelope and SVPs are difficult to distinguish by IP. To address the effect ERGIC53 and Sec24A on filament SVP secretion, the authors need to include the construct expressing LMS (the construct from C. Sureau) as the control for IP and include the immunoblot results to show the pull-down efficiency. The panel 8B(left) only shows intracellular levels (i.e., input of IP), which is not helpful in addressing the IP efficiency. Thus, adding the experiments for demonstrating IP-efficiency will strengthen that ERGIC53 and Sec24A affect intracellular viral transport.
The IP efficiency. Yes, we doubtlessly agree that the L-specific IP will pull-down both, filament SVPs and HBV virions, and even more filaments as compared to virions. However, we feel that this does not impact the outcome of the results, shown in Figure 8B, because our read-out is the PCR measurement of progeny HBV genomes. There is concordant evidence that filaments are devoid of capsids/NCs and hence of encapsidated viral genomes. Accordingly, if the L-specific IP will bring down filaments, this is unlikely to affect the PCR results. But as requested, we have now examined the IP efficiency. This is now shown in a new Figure 8C and described in the text (lines 583-587) which demonstrates effective and comparable IP efficiencies. In addition, we performed a similar experiment with the LMS-expressing construct, as suggested. Thereby, we did not observed significant differences in IP efficiencies between the HBV replicon and LMS-expression construct. Out of this reason – and to avoid to lengthen our manuscript to much – we did not include these results. But do convince our reviewer, these results are included herein (Please see the attached PDF-file).
We appreciate the comment: ”Whether knock-down of ERGIC53 or Sec24A affects filament SVPs”, as it is very reasonable. This is one subject of our ongoing research, but thus far we do not have clear answers. Unfortunately, there is no robust expression system that allows a sole formation and secretion of HBV filament SVPs. Jiang et al. (J. Virol. 2015, Ref. # 23) studied filament release in HepG2 and HuH-7 cells by using a construct expressing LMS under control of their natural promotors in the absence of an ongoing replication (a construct that is comparable to the construct of C. Sureau). The authors succeeded to detect filaments in the supernatants of transfected cells, as shown by WB and electron microscopic analyses. However, “in these specimens a lot of spheres 22 nm in diameter were observed as well”, indicating that the usage of an LMS construct does not guarantee exclusive filament formation. In future work, we attempt to optimize an LMS expression construct (by changing the ratios of the three envelope proteins) with the aim to primarily produce filaments.
Reviewer #1: Minor points
- Line 12: “characterization of intersections…” should be “characterization of interactions…”.
This has been changed (line 12).
- Line 30-31: “About 240 million people are persistently infected, …” should use the most recent data from WHO, which is about 257 million.
This has been changed (line 32).
- Line 45-46: “This replication intermediate then serves as a template for RNA transcription and translation into the viral proteins in the cytoplasm.” can be rephrased into “This replication intermediate serves as the template for viral RNA transcription.”
This has been rephrased (line 49).
- Line 84: “The viral S envelope protein alone is sufficient for the production of SVPs that are formed by…” should precisely use spherical SVPs.
This has been changed (line 87-88).
- Line 87: “spherical SVPs does neither require N146-linked N-glycans” should be “spherical SVPs does neither require N146-linked glycans”.
This has been changed (line 91).
- Line 106: “…recognizes the viral envelope in a N146-glycan-dependent manner…” should be “…recognizes the viral envelope in an N146-linked glycan-dependent manner”.
This has been changed (line 113-114).
- Line 274: “Unlike those of HBV subviral envelope particles (SVPs), intracellular trafficking pathways of the viral envelope are poorly understood.” This sentence is unclear and could be rephrased into “Unlike those of HBV spherical subviral envelope particles (SVPs) formed by S alone, intracellular trafficking pathways of the L is poorly understood.” See the major comments above.
This has been modified (lines 282-283).
- Line 363: “(B) HuH-7 cells were transfected with HBV.Cminus… “ should be “(G) HuH-7 cells were transfected with HBV.Cminus…”.
This has been changed (now H) (line 379).
- Line 384: delete the nucleocapsids and use NCs.
This has been changed (line 399).
- Line 420: “is transported out of the ER to the ERGIC in control cells (Figure 4A).” should be “is transported out of the ER to the ERGIC in control cells (Figure 4B)”.
This has been changed (line 435).
- Line 422: “…that no longer coincided with ERGIC-53 signals (Figure 4A).” should be “that no longer coincided with ERGIC-53 signals (Figure 4B).”
This has been changed (line 437).
- Line 440-441: “In contrast, the HBV viral envelope could not be traced within the ERGIC as no colocalization between L and ERGIC-53 was detectable in control cells (Figure 4B).” should be “In contrast, the HBV Large envelope (or L) protein could not be traced within the ERGIC as no colocalization between L and ERGIC-53 was detectable in control cells (Figure 4C).”
This has been changed (line 456).
- In Line532, authors wrote transfected with wt or Env.N146D replicons, but in Fig.7B labeled as the HBV.Envminus+ Env.N146D, which means co-transfection. Please correct it properly.
This has been corrected (lines 547-548).
- Line 717-720: “There, ERGIC-53 and the COPII coat proteins presumably come into play, but their temporal and spatial mode of action is yet ambiguous. In particular, we still do not know whether Sec24A may act as a direct COPII adaptor for the viral envelope or as an indirect adaptor, recognizing the envelope-associated ERGIC-53 cargo receptor.” should be “Therefore, ERGIC-53 and the COPII coat proteins presumably come into play. However, their temporal and spatial mode of action is yet ambiguous. In particular, we still do not know whether Sec24A acts as a direct COPII adaptor for the viral envelope or as an indirect adaptor, recognizing the envelope-associated ERGIC-53 cargo receptor.”
This has been changed (lines 744-747).
Additional changes by the authors:
During proof-reading, we noted that the sequences of the oligonucleotides used for mutagenesis of Sec24A were misstated. This has been now corrected (line 153-154).

Reviewer 2 Report
Zeyen and colleagues examined the role of COPII protein complex during HBV trafficking and egress. Here, they have shown that the HBV assembly starts at the ER and implicated ER-Golgi intermediate compartment 53 protein (ERGIC-53) as important protein interacting directly with HBV envelope via N146-glycan. Despite lacking experiments with true infectious HBV model, they present clear and visual evidence supporting their conclusions. There are only few minor points that has to be addressed.
Minor points:
Line 363: change (B) to (G)
Line 375: specify which cells in “cells were treated….”
Line 392: use different word instead of significant (distinct, obvious,…)
Line 404: in the legend is β-actin and in the figure is Tubulin
Chapter 3.4 Check labeling of figure 4 in the text. A should be B, and line 441 Figure 4B should be 4C, refer correctly to 4A in the text
Figure 5C WB: Were two siERGIC-53 used or it is duplicated experiment?
Line 574: delete “Treatment of”
Author Response
Reviewer #2: Minor points
Line 363: change (B) to (G)
Thanks, this has been changed (now to “H”) (line 379).
Line 375: specify which cells in “cells were treated….”
This has been changed (HuH-7 cells) (line 356).
Line 392: use different word instead of s90ificant (distinct, obvious,…)
This has been changed (obvious) (line 407).
Line 404: in the legend is β-actin and in the figure is Tubulin
Thanks, this has been changed (line 419).
Chapter 3.4 Check labeling of figure 4 in the text. A should be B, and line 441 Figure 4B should be 4C, refer correctly to 4A in the text
This has been corrected (lines 431, 435, 437, 443).
Figure 5C WB: Were two siERGIC-53 used or it is duplicated experiment?
We now indicated in the revised manuscript that this experiment was done in duplicate. In addition, we would like to stress that the used siRNA against ERGIC-53 had been approved by independent labs in different cell lines, as already stated in our original manuscript (lines 507-508)
Line 574: delete “Treatment of”
Thanks, this has been deleted (line 594).

Round 2
Reviewer 1 Report
The authors have addressed my comments very well.